# Uncertainty Estimation for Multi-view Data: The Power of Seeing the Whole Picture

**Myong Chol Jung**
Monash University
david.jung@monash.edu

**He Zhao**
CSIRO's Data61
he.zhao@ieee.org

**Joanna Dipnall**
Monash University
jo.dipnall@monash.edu

**Belinda Gabbe**
Monash University
belinda.gabbe@monash.edu

**Lan Du**[*]
Monash University
lan.du@monash.edu

## Abstract

Uncertainty estimation is essential to make neural networks trustworthy in real-world applications. Extensive research efforts have been made to quantify and reduce predictive uncertainty. However, most existing works are designed for unimodal data, whereas multi-view uncertainty estimation has not been sufficiently investigated. Therefore, we propose a new multi-view classification framework for better uncertainty estimation and out-of-domain sample detection, where we associate each view with an uncertainty-aware classifier and combine the predictions of all the views in a principled way. The experimental results with real-world datasets demonstrate that our proposed approach is an accurate, reliable, and well-calibrated classifier, which predominantly outperforms the multi-view baselines tested in terms of expected calibration error, robustness to noise, and accuracy for the in-domain sample classification and the out-of-domain sample detection tasks[2].

## 1 Introduction

Reliable uncertainty estimation is critical for deploying deep learning models in a number of domains such as medical imaging diagnosis [35] or autonomous driving [8]. Even with accurate predictions, domain experts still raise questions of how trustworthy the models are [39]. For example, when a model's prediction contradicts a domain expert's opinion, the quantification of the uncertainty of the model's predictions can help determine model's reliability and justify model use.

Recently, uncertainty estimation of neural networks has been an active research area, where many methods of quantifying uncertainty in predictions have been proposed [44, 9, 40, 50, 32, 24]. The majority of existing work focuses on uncertainty estimation for unimodal data. However, in many practical problems, data can exhibit in multi-views or multi-modalities. For example, LiDAR, radar, and RGB cameras can simultaneously capture complementary information about a scene [49], and computed tomography (CT) scans and x-ray images can be analyzed together to diagnose a disease [3]. Trustworthy uncertainty estimation with multi-view or multi-modalities data is important because the challenges it faces may differ from a unimodal setting (e.g., maintaining accurate predictions with one of the input views' domain shifted). Despite the success of existing work on unimodality, modelling and estimating uncertainty for multi-view data remain a less explored question [11].

---

[*]Corresponding author
[2]We provide our code at `https://github.com/davidmcjung/multiview_uncertainty_estimation`

36th Conference on Neural Information Processing Systems (NeurIPS 2022).

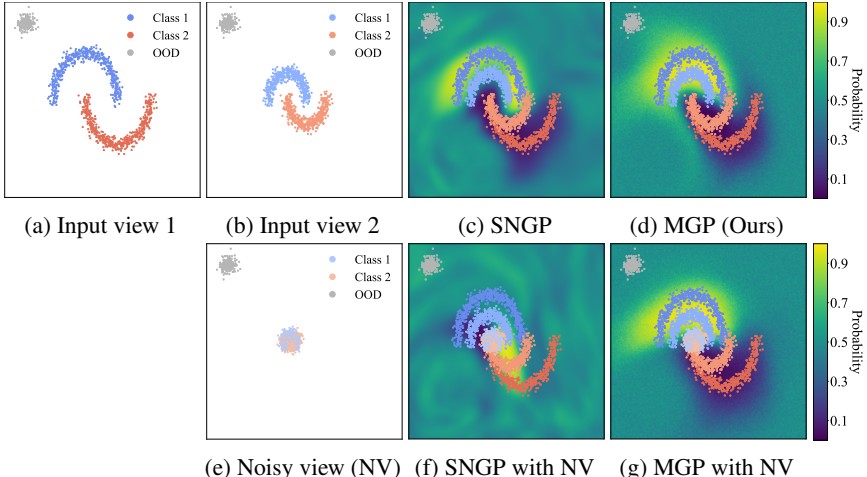

Figure 1: Visualization on a synthetic multi-view moon dataset. Top row: the dataset has two views and two classes (e.g., blue upper circles in (a) and (b) are two views of Class 1), and an OOD class (grey); (c) and (d) are the predictive probability surfaces of SNGP and our MGP. Bottom row: A new noisy view (e) is added to the data; (f) and (g) are the predictive probability surfaces of SNGP and our MGP with the noisy view. The darker the region is (i.e., dark blue), the lower the probability of being class 1. Since SNGP is a unimodal model, input views are fused into a unimodal dataset. The difference between (c) and (f) shows SNGP cannot correctly capture the input shape in the presence of noise. MGP, however, is robust to noise, shown by minimal difference between (d) and (g).

A way of solving this problem is to fuse multi-modalities into one modality and directly apply existing unimodal methods. However, even the state-of-the-art unimodal model (e.g., SNGP [29]) can be prone to noise if one of the views in a multi-view dataset is noisy, as shown in Figure 1. Without the noisy view, unimodal models can produce accurate and confident predictions nearby the training domain. However, with the noisy view, the predictions become uncertain for samples even close to the training domain (see Figure 3). We also show that the existing multi-view classifiers (e.g., TMC [11]) have limited capacity to detect out-of-domain (OOD) samples in our experiments (see Table 4).

To this end, we propose the Multi-view Gaussian Process (MGP) that is a tailored framework providing intrinsic uncertainty estimation for classification of multi-view/modal data. Specifically, MGP consists of a dedicated Gaussian process (GP) expert for each view whose predictions are aggregated by the product of expert (PoE). In our proposed method, there is a natural way of capturing uncertainty by measuring the distance between training set and test samples in the reproducing kernel Hilbert space (RKHS). The contributions of our method can be summarized as follows:

1. We propose a new uncertainty estimation framework with GPs for multi-view data, which is an under-explored yet increasingly important problem in safety-critical applications.
2. The framework provides better uncertainty estimation through a product of expert model, providing more robustness in dealing with noise and better capacity of detecting OOD data.
3. We develop an effective variational inference algorithm to approximate multi-view posterior distributions in a principled way.
4. We conduct comprehensive and extensive experiments on both synthetic and real-world data, which show that our method achieves the state-of-the-art performance for uncertainty estimation of multi-view/modal data.

## 2 Muti-view Gaussian Process

Given training data $\boldsymbol{X} = \{\boldsymbol{X}_1, \boldsymbol{X}_2, ..., \boldsymbol{X}_V\}$ where $V$ is the number of views, each view consists of a training set of $N$ samples $\boldsymbol{X}_v = \{\boldsymbol{x}_{v,i}\}_{i=1}^N$ and labels $\boldsymbol{y} = \{y_i\}_{i=1}^N$. In other words, the $i^{\text{th}}$ data sample consists of $V$ views $\{\boldsymbol{x}_{v,i}\}_{v=1}^V$ (e.g., the CUB dataset consists of images as the first view and captions as the second view) and $y_i$ is the data sample's ground-truth label shared across the views. Without loss of generality, a multiview classification or regression problem can be formulated as

predicting $\boldsymbol{y}_*$ given testing samples $\{\boldsymbol{X}_{*,v}\}_{v=1}^V$. In this paper, we propose the Multi-view Gaussian Process (MGP), a novel framework for multi-view/modal data, where in a nutshell we first apply a GP to each view of the data and then combine in a principled way the predictions from all the GPs as a unified prediction by using the product of expert (PoE) [28].

## 2.1 GP for an Individual View

For each view, we consider a multiclass classification problem with $C$ classes. We set $C$ independent Gaussian priors over latent function $\boldsymbol{f}_v(\cdot)$ with zero-mean and $N \times N$ covariance matrix $\boldsymbol{K}_{NN}$ whose element is $\boldsymbol{K}_{ij} = k(\boldsymbol{x}_{v,i}, \boldsymbol{x}_{v,j})$, where $k(\cdot, \cdot)$ is a kernel function. The radial basis function (RBF) which is commonly used in the GP literature [51, 33] is selected in this paper as the covariance function. It is defined as $k(\boldsymbol{x}, \boldsymbol{x}') = \sigma_v^2 \exp\left(\frac{-(\boldsymbol{x}-\boldsymbol{x}')^2}{2l_v^2}\right)$, where $\sigma_v^2$ is the signal variance and $l_v$ is the length-scale for each GP which are parameters to be optimized.

To bypass the limitations of standard GPs [28], namely high computational cost $\mathcal{O}(N^3)$ and inconvenience of applying stochastic gradient descent (SGD), we propose to leverage the sparse variational GP (SVGP) [28, 14, 15, 43], which is detailed as follows. With SVGP, we introduce $M$ ($M < N$) inducing points $\boldsymbol{Z}_v$ representing the training samples of view $v$ with a smaller number of points, and the inducing variable $\boldsymbol{u}_v$ is the latent function evaluated at the inducing points (i.e., $\boldsymbol{u}_v = \boldsymbol{f}_v(\boldsymbol{Z}_v)$) where both $\boldsymbol{Z}_v$ and $\boldsymbol{u}_v$ are random variables to be optimized. Similar to the Gaussian prior set for the latent function, a joint prior can be set as:

$$\begin{bmatrix} \boldsymbol{f}_v \\ \boldsymbol{u}_v \end{bmatrix} \sim \mathcal{N}\left(\boldsymbol{0}, \begin{bmatrix} \boldsymbol{K}_{NN} & \boldsymbol{K}_{NM} \\ \boldsymbol{K}_{NM}^{\mathrm{T}} & \boldsymbol{K}_{MM} \end{bmatrix}\right) \tag{1}$$

where we use $\boldsymbol{f}_v$ to indicate $\boldsymbol{f}_v(\boldsymbol{X}_v)$ for notation convenience. The use of inducing points can reduce the computational cost to $\mathcal{O}(M^3)$ [28]. We outline the likelihood of GPs in Section 2.2 and the posterior in Section 2.3.

## 2.2 GPs for Multi-view Data with PoE

**Product of Experts (PoE)** With one GP expert for each view, we propose to combine the GP experts into a unified prediction by using the PoE mechanism [17, 28, 7, 6]. Specifically, we aggregate posterior distributions of individual views by:

$$p(\boldsymbol{f}|\boldsymbol{X}, \boldsymbol{y}) \propto \prod_v p(\boldsymbol{f}_v|\boldsymbol{y}) \tag{2}$$

For Gaussian posteriors with mean $\boldsymbol{\mu}_v$ and covariance $\boldsymbol{\Sigma}_v$, the aggregation using Equation (2) forms the unified posterior's mean and covariance expressed as:

$$\boldsymbol{\mu} = \left(\sum_v \boldsymbol{\mu}_v \boldsymbol{\Sigma}_v^{-1}\right) \boldsymbol{\Sigma}, \quad \boldsymbol{\Sigma} = \left(\sum_v \boldsymbol{\Sigma}_v^{-1}\right)^{-1} \tag{3}$$

**Dirichlet-based Likelihood** In order to apply Equation (3) to a multi-view problem, the latent function $\boldsymbol{f}_v$ in each view should refer to the same observable variable (i.e., $\mathcal{N}(\boldsymbol{a}|b,c)$ cannot be combined with $\mathcal{N}(\boldsymbol{c}|d,e)$ for $\boldsymbol{a} \neq \boldsymbol{c}$). However, in GP classification, the latent function is a non-observable *nuisance function* that is squashed through sigmoid or softmax function to estimate labels [51], which is not necessarily the same for every independent view. We alleviate this problem by reparameterizing the class labels to regression labels by:

$$\widetilde{\boldsymbol{y}}_i = \boldsymbol{f}_v(\boldsymbol{x}_{v,i}) + \boldsymbol{\epsilon}, \quad \boldsymbol{\epsilon} \sim \mathcal{N}(\boldsymbol{0}, \tilde{\boldsymbol{\sigma}}_i^2)$$

where $\widetilde{\boldsymbol{y}}_i$ is the transformed label and $\tilde{\boldsymbol{\sigma}}_i^2$ is the noise parameter fixed for all views. Since $\widetilde{\boldsymbol{y}}_i$ and $\tilde{\boldsymbol{\sigma}}_i^2$ are shared across the views, we ensure that $\boldsymbol{f}_v(\boldsymbol{x}_{v,i})$ refers to the same variable. By using the log-normal distribution, the Gaussian likelihood can be used in the log space as $p(\widetilde{\boldsymbol{y}}_i|\boldsymbol{f}_v) = \mathcal{N}(\boldsymbol{f}_v, \tilde{\boldsymbol{\sigma}}_i^2)$.

To transform the class labels to regression labels, we propose to adopt representing the class probability $\boldsymbol{\pi}_i = [\pi_{i,1}, \pi_{i,2}, ..., \pi_{i,C}]$ over a Dirichlet distribution with the categorical likelihood [33]:

$$p(y_i|\boldsymbol{\alpha}_i) = \mathrm{Cat}(\boldsymbol{\pi}_i), \quad \text{where} \quad \boldsymbol{\pi}_i \sim \mathrm{Dir}(\boldsymbol{\alpha}_i)$$

$$\pi_{i,c} = \frac{g_{i,c}}{\sum_{j=1}^C g_{j,c}}, \quad \text{where} \quad g_{i,c} \sim \mathrm{Gamma}(\alpha_{i,c}, 1) \tag{4}$$

where $\boldsymbol{\alpha}_i = [\alpha_{i,1}, \alpha_{i,2}, ..., \alpha_{i,C}]$ is the concentration parameters, the shape parameter for Gamma distribution is $\alpha_{i,c}$, and the scale parameter for Gamma distribution is $\theta = 1$. We approximate the Gamma distribution in (4) with $\tilde{g}_{i,c} \sim \text{Lognormal}(\tilde{y}_{i,c}, \tilde{\sigma}_{i,c}^2)$ by moment matching:

$$\alpha_{i,c} = \exp\left(\tilde{y}_{i,c} + \tilde{\sigma}_{i,c}^2/2\right), \quad \alpha_{i,c} = \left(\exp\left(\tilde{\sigma}_{i,c}^2\right) - 1\right)\exp\left(2\tilde{y}_{i,c} + \tilde{\sigma}_{i,c}^2\right)$$

Thus, the transformed labels and the noisy parameter are expressed in terms of the concentration parameters:

$$\tilde{\sigma}_{i,c}^2 = \log\left(1/\alpha_{i,c} + 1\right), \quad \tilde{y}_{i,c} = \log \alpha_{i,c} - \tilde{\sigma}_{i,c}^2/2 \tag{5}$$

where $\alpha_{i,c} = 1 + \alpha_\epsilon$ if $y_{i,c} = 1$ and $\alpha_{i,c} = \alpha_\epsilon$ if $y_{i,c} = 0$ with the one-hot label $y_{i,c}$. $\alpha_\epsilon$ is a parameter to prevent the noise parameter from converging to infinity. See Appendix for the impacts of $\alpha_\epsilon$ on the model performance. Compared with other transforming methods such as Platt scaling [38], the used Dirichlet likelihood compromises classification accuracy less and requires no post-hoc calibrations after training.

## 2.3 Training of the Proposed Framework

Given the priors from Section 2.1 and the Gaussian likelihood from Section 2.2, the goal of training our framework is to estimate a posterior distribution via variational inference (VI) [14, 15, 4]. By using Equation (2), we propose an aggregated variational distribution for all the views as:

$$q_{PoE}(\boldsymbol{f}) \propto \prod_v q(\boldsymbol{f}_v) \tag{6}$$

where $q(\boldsymbol{f}_v)$ is the variational distribution for each view that approximates the true posterior. We define $q(\boldsymbol{f}_v)$ as:

$$q(\boldsymbol{f}_v) := \int p\left(\boldsymbol{f}_v | \boldsymbol{u}_v\right) q(\boldsymbol{u}_v)\, d\boldsymbol{u}_v. \tag{7}$$

where $p\left(\boldsymbol{f}_v | \boldsymbol{u}_v\right)$ is the conditional prior from Equation (1), and $q(\boldsymbol{u}_v)$ is the marginal variational distribution of $\mathcal{N}(\boldsymbol{m}_v, \boldsymbol{S}_v)$ with optimizable model parameters $\boldsymbol{m}_v$ and $\boldsymbol{S}_v$. The analytical solution of (7) is provided in Appendix. VI seeks to minimize the following Kullback–Leibler divergence (KL) between the true posterior and variational distributions:

$$\text{KL}\left[q_{PoE}(\boldsymbol{f}) || p(\boldsymbol{f}|\boldsymbol{X}, \widetilde{\boldsymbol{y}}_c)\right] \tag{8}$$

where $\widetilde{\boldsymbol{y}}_c = \{\tilde{y}_{i,c}\}_{i=1}^N$.

**Lemma 1** (Additive Property of KL Divergence). *If* $x = [x_1, \cdots, x_n] \in \mathcal{X}$, $p(x) = \prod_i^n p(x_i)$ *and* $q(x) = \prod_i^n q(x_i)$, *we have:*

$$\text{KL}\left[p(x) || q(x)\right] = \sum_i^n \text{KL}\left[p(x_i) || q(x_i)\right] \tag{9}$$

**Theorem 2** (KL Divergence with PoE). *With Equations* (2) *and* (6), *we have:*

$$\text{KL}\left[q_{PoE}(\boldsymbol{f}) || p(\boldsymbol{f}|\boldsymbol{X}, \widetilde{\boldsymbol{y}}_c)\right] = \sum_v \text{KL}\left[q(\boldsymbol{f}_v) || p(\boldsymbol{f}_v | \widetilde{\boldsymbol{y}}_c)\right] \tag{10}$$

According to Theorem 2, the VI for the PoE splits to the VI of each expert/view. For the $v^{\text{th}}$ view, the VI minimizes $\text{KL}\left[q(\boldsymbol{f}_v) || p(\boldsymbol{f}_v | \widetilde{\boldsymbol{y}}_c)\right]$, which can be turned into the maximization of the evidence lower bound (ELBO):

$$\text{ELBO}_v = \sum_{i=1}^N \mathbb{E}_{q(\boldsymbol{f}_{v,i})}\left[\log p(\tilde{y}_{i,c}|\boldsymbol{f}_{v,i})\right] - \beta \cdot \text{KL}\left[q(\boldsymbol{u}_v) || p(\boldsymbol{u}_v)\right] \tag{11}$$

where $\beta$ is a parameter to control the KL term, similar to [16], which can be interpreted as a regularization term. Proofs of Equation (9)-(11) are provided in Appendix.

In order to apply SGD, we match the expectation of stochastic gradient of the expected log likelihood term to the full gradient by multiplying the number of batches to the log likelihood term in Equation (11) [15]. The overall loss for all experts is:

$$\mathcal{L} = -\sum_{v=1}^V \text{ELBO}_v \tag{12}$$

The training steps are summarized in Algorithm 1.

**Algorithm 1:** Learning MGP

---

**Input:** $V$ views of training data
$\quad\quad\boldsymbol{X} = \{\boldsymbol{X}_v\}_{v=1}^V$ where each view
$\quad\quad$ has $N$ samples of $\boldsymbol{X}_v = \{\boldsymbol{x}_{v,i}\}_{i=1}^N$
$\quad\quad$ and $\boldsymbol{y} = \{y_i\}_{i=1}^N$.
**Transform:** Reparameterize $\widetilde{\boldsymbol{y}}_c$ by (5)

1 **for** *minibatch* **do**
2 $\quad$ **for** $v = 1$ **to** $V$ **do**
3 $\quad\quad$ Compute $q(\boldsymbol{f}_v)$ by (7)
4 $\quad\quad$ Calculate $\text{ELBO}_v$ by (11)
5 $\quad$ **end for**
6 $\quad$ Sum ELBOs by (12)
7 $\quad$ SGD update $\{l_v, \sigma_v^2, \boldsymbol{Z}_v, \boldsymbol{m}_v, \boldsymbol{S}_v\}_{v=1}^V$
8 **end for**

---

**Algorithm 2:** Inference of MGP

---

**Input:** $V$ views of testing data
$\quad\quad\boldsymbol{X}_* = \{\boldsymbol{X}_{*,v}\}_{v=1}^V$

1 **for** $v = 1$ **to** $V$ **do**
2 $\quad$ Compute $q(\boldsymbol{f}_{*,v})$ by (13)
3 $\quad$ Calculate $\gamma(\boldsymbol{X}_{*,v})$ by (17)
4 **end for**
5 Aggregate $q_{PoE}(\boldsymbol{f}_*)$ by (16)
**Output:** $\mathbb{E}\left[\pi_{i,c}\right]$ and $\mathbb{V}\left[\pi_{i,c}\right]$ of class
$\quad\quad\quad$ probability by (14)

---

## 2.4 Inference on Test Samples

Given test samples $\boldsymbol{X}_* = \{\boldsymbol{X}_{*,v}\}_{v=1}^V$, the predictive distribution $p(\boldsymbol{f}_{*,v}|\widetilde{\boldsymbol{y}}_c)$ is estimated by the varitional distribution as:

$$p(\boldsymbol{f}_{*,v}|\widetilde{\boldsymbol{y}}_c) \approx q(\boldsymbol{f}_{*,v}) = \int p(\boldsymbol{f}_{*,v}|\boldsymbol{u}_v)q(\boldsymbol{u}_v)\,d\boldsymbol{u}_v \tag{13}$$

where $p(\boldsymbol{f}_{*,v}|\boldsymbol{u}_v)$ can be formed by the joint prior distribution similar to Equation (1) (see Appendix for a full derivation). Similar to Equation (6), we aggregate the predictive distributions to form $q_{PoE}(\boldsymbol{f}_*)$ that is sampled to approximate Gamma-distributed samples which in the end form the posterior of Dirichlet distribution as follows:

$$\mathbb{E}\left[\pi_{i,c}\right] = \int \frac{\exp\left(f_{i,c,*}\right)}{\sum_j \exp\left(f_{i,j,*}\right)} q_{PoE}(f_{i,c,*})\,d\boldsymbol{f}_*$$

$$\mathbb{V}\left[\pi_{i,c}\right] = \int \left(\frac{\exp\left(f_{i,c,*}\right)}{\sum_j \exp\left(f_{i,j,*}\right)} - \mathbb{E}\left[\pi_{i,c}\right]\right)^2 q_{PoE}(f_{i,c,*})\,d\boldsymbol{f}_* \tag{14}$$

Equation (14) can be approximated with the Monte Carlo method. See Appendix for the impacts of the number of Monte Carlo samples on classification performance and inference time. The aggregated predictive distribution can also be weighted by each expert's predictive distribution by:

$$q_{PoE}(\boldsymbol{f}_*) \propto \prod_v \left(q(\boldsymbol{f}_{*,v})\right)^{\gamma(\boldsymbol{X}_{*,v})} \tag{15}$$

where $\gamma(\boldsymbol{X}_{*,v})$ is the weight controlling the influence of each expert to the aggregated prediction. The mean and covariance of $q_{PoE}(\boldsymbol{f}_*)$ with $\gamma(\boldsymbol{X}_{*,v})$ are:

$$\boldsymbol{\mu}_W = \left(\sum_v \boldsymbol{\mu}_v \gamma(\boldsymbol{X}_{*,v})\boldsymbol{\Sigma}_v^{-1}\right)\boldsymbol{\Sigma}_W, \quad \boldsymbol{\Sigma}_W = \left(\sum_v \gamma(\boldsymbol{X}_{*,v})\boldsymbol{\Sigma}_v^{-1}\right)^{-1} \tag{16}$$

In our experiments, we use negative entropy of predictive distribution:

$$\gamma(\boldsymbol{X}_{*,v}) = -H(q(\boldsymbol{f}_{*,v})) \tag{17}$$

Note that the original PoE [17] in Equation (6) is recovered if $\gamma(\boldsymbol{X}_{*,v}) = 1$. The intuition behind choosing negative entropy is that the experts with lower posterior entropy, which means the lower uncertainty, gain more contribution to the aggregated predictions. Please note that other choices of $\gamma(\boldsymbol{X}_{*,v})$ can also be applied such as the difference in entropy from prior to posterior [6] and negative predictive variance [7]. We obtain the better empirical results with negative entropy, but the choice of function is flexible. The inference steps are summarized in Algorithm 2.

# 3 Related Work

**Uncertainty Estimation with GP** GP has been one of the gold standards of uncertainty estimation because of GP's high sensitivity to domain shift. One of the common ways to implement GP with deep learning models is to place GP at the output layer on top of extracted features. The features are often extracted from deterministic deep neural networks [46, 29, 5], Bayesian neural networks [26], or graph data [30]. Similarly, MGP builds on these approaches and can be combined with various feature extractors. However, our method differs from all of above studies because these studies are designed for unimodal data. However, MGP is a multi-view GP. Other variants utilizing kernel learning for uncertainty estimation include deep GP [27] and RBF network [45].

**Multi-view Learning** Multi-view and multimodal learning aim for various downstream tasks by leveraging multiple data sources that describe the same event or object. Canonical correlation analysis (CCA) [18] holds a long history, which finds a common representation of multiple sources [1, 47]. Similarly, contrastive learning builds the common representation by forming positive pairs and negative pairs [42, 12]. Also, view-specific representations are learned to make models robust to missing input view [52]. Recently, it has been theoretically and empirically shown that vision-language models outperform unimodal models [19, 25, 21]. Other methods include gradient-blending [48] and hierarchical metric learning [53]. Despite these extensive studies in multi-view and multimodal learning, most of them are not mainly designed for uncertainty estimation.

**Multi-view Uncertainty Estimation** Few studies have been designed and evaluated for multi-view and multimodal uncertainty estimation. Multimodal regression with mixture of normal-inverse Gamma distributions yields promising uncertainty estimation and predictions with the real-world data [31]. However, this method is designed for regression, which differs from our method for classification. The closest study to ours is the trusted multi-view classifier (TMC) which combines evidence from different views by using Dempster's combination rule [11]. However, the Dempsterd's combination rule ignores predictions of conflicting views, which is an undesirable property especially in high-risk applications [11, 20]. In addition, our experiments show that TMC is overconfident about the OOD samples.

# 4 Experiments

## 4.1 Synthetic Dataset

To illustrate predictive behaviours of baselines and MGP, we constructed a multi-view synthetic dataset with the Scikit-learn's moon dataset [3] by scaling the data with three different scaling factors (the same dataset used in Figure 1). Each view consists of 1,000 data points formed as two half circles: upper circle (class 1) and lower circle (class 2). In each view, data points share the same relative locations with the same labels. Note that the points in the third view overlap each other, representing a noisy view.

Deep Ensemble [23] with late fusion [2] (DE(LF)), TMC, and MGP are comprised of dedicated classifiers for every view (view 1, 2, and 3 in Figure 2), and the predictions made in the views are combined as single prediction represented as multi-view in Figure 2. Since SNGP[4] is a single-view classifier, data points of all the views were concatenated as input.The results of SNGP are shown in Figure 1. For experimental details, see Appendix.

One of the benefits of having a multi-view uncertainty estimator is that the prediction accuracy of an ideal multi-view classifier remains high even if one of the input views does not provide meaningful information. This feature can be achieved by assigning lower weights to the views with high uncertainties when combining predictions. High uncertainty here refers to unconfident predictions with a uniform class probability across classes.

Figure 2 shows that the combined predictions of DE(LF) are moderately affected by the noise because the predictions are averaged across all the views. TMC and MGP, on the other hand, are not affected

---

[3]`https://scikit-learn.org/stable/modules/generated/sklearn.datasets.make_moons.html`

[4]To provide a fair comparison, the same feature extractor without spectral normalization is used for DE(LF), TMC, MGP, and SNGP. See Appendix for results of SNGP with spectral normalization.

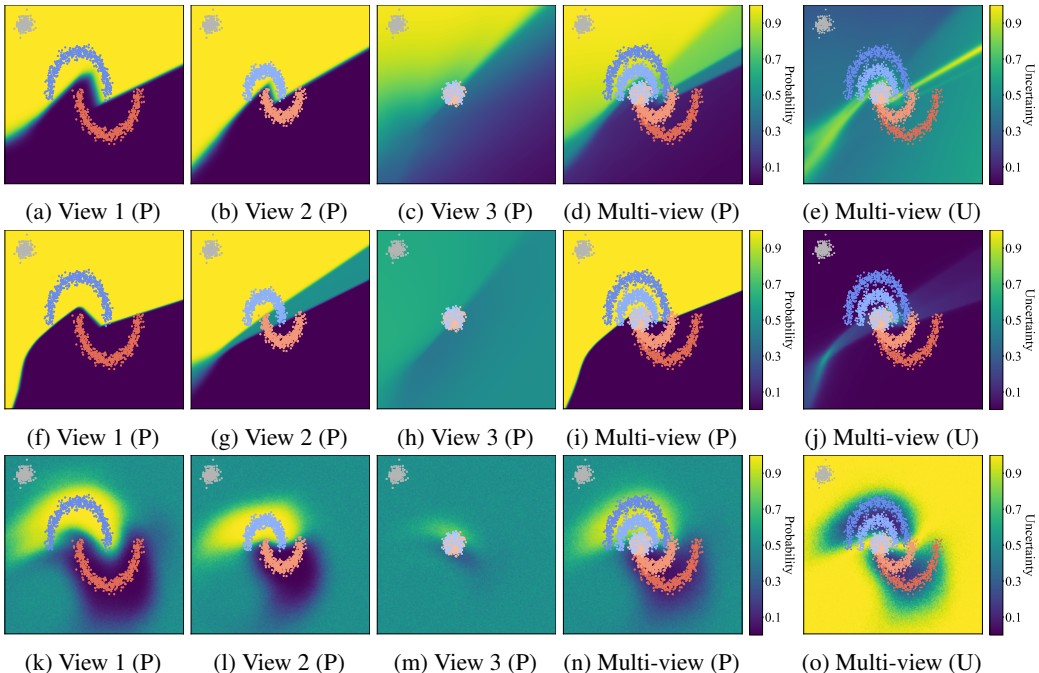

|  | (a) View 1 (P) | (b) View 2 (P) | (c) View 3 (P) | (d) Multi-view (P) | (e) Multi-view (U) |
|  | (f) View 1 (P) | (g) View 2 (P) | (h) View 3 (P) | (i) Multi-view (P) | (j) Multi-view (U) |
|  | (k) View 1 (P) | (l) View 2 (P) | (m) View 3 (P) | (n) Multi-view (P) | (o) Multi-view (U) |

Figure 2: Predictive probability surface (P) of class 1 and uncertainty estimation surface (U) from DE(LF) (a)-(e), TMC (f)-(j), and MGP (k)-(o) trained with upper circle (class 1) and lower circle (class 2) synthetic multi-view dataset. Grey data points are OOD samples. View 3 is intentionally made to be noisy. Uncertainty surfaces for individual views are plotted in Appendix.

by the noisy view (View 3). The majority of areas in the noisy view show high uncertainty (i.e., class probability close to 0.5) which have minimal impact on combining predictions because TMC and MGP are aware of uncertainty of each view. However, SNGP's prediction is heavily impacted by the noisy view as shown in Figure 1c. If SNGP were trained without any noisy view, it can properly estimate class probability (Figure 1f). This difference between Figure 1c and 1f illustrates the degrading effect of noisy view to single-view classifiers.

Although TMC is robust to noise, it produces overconfident predictions at the regions far from decision boundaries (see Figure 2f-2i). As a result, OOD samples are indistinguishable from in-domain samples (see Figure 2j). This is mainly caused by lack of distance awareness which MGP and SNGP have in common due to GP [29]. Figure 2k-2o show that MGP is well aware of the distance between the training domain and OOD by allocating high uncertainty at the OOD samples where uncertainty is calculated by the sum of variance across all classes.

## 4.2 Robustness to Noise

**Experimental Settings** We used six multi-view datasets [11] (Handwritten, CUB, PIE, Caltech101, Scene15, and HMDB) with train-test split of 0.8:0.2. For testing robustness to noise, we added Gaussian noise of zero mean to half of the views for each dataset, following the experimental setting in [11]. In order to have the same impact of noise to all views, we normalized each view first and then added the noise because the range of raw data of each view varies significantly.[5] We increased the noise standard deviation from 0.01 to 10. To report test results invariant to selecting which views to add the noise, we ran all combinations of selecting noisy views (i.e., $\binom{V}{V/2}$ configurations) and report its average for each noise level.

**Compared Methods** We selected three single-view baselines and two multi-view baselines. For single-view baselines, we used early fusion (EF) technique [2] by concatenating multi-view features

---

[5]TMC's authors added the noise first and then normalized the noisy input. The results of this setting are reported in Appendix.

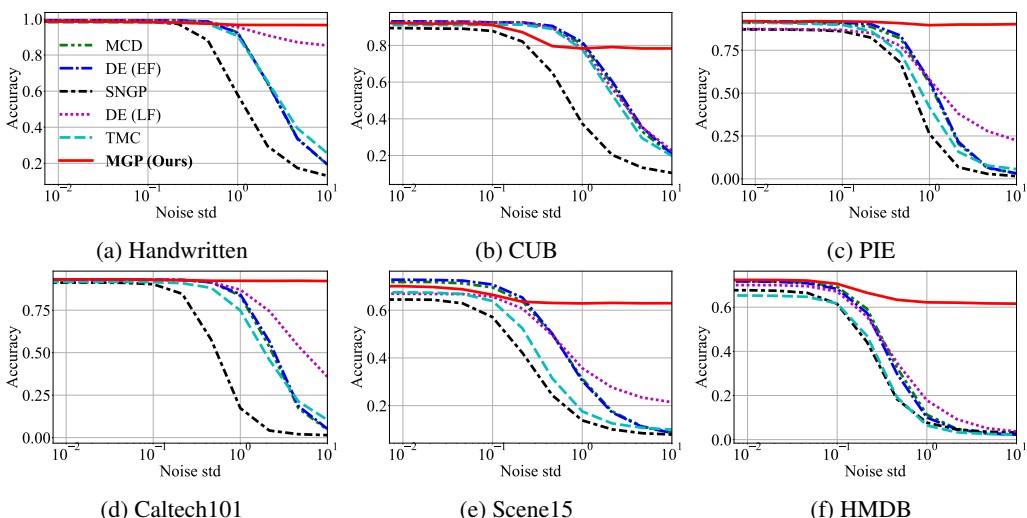

| | | | | | |
|---|---|---|---|---|---|
| (a) Handwritten | | (b) CUB | | (c) PIE | |
| (d) Caltech101 | | (e) Scene15 | | (f) HMDB | |

Figure 3: Domain-shift test accuracy where Gaussian noise is added to half of the views.

Table 1: In-domain test accuracy ↑

| | Dataset | | | | | |
|---|---|---|---|---|---|---|
| Method | Handwritten | CUB | PIE | Caltech101 | Scene15 | HMDB |
| MC Dropout | **99.25±0.00** | 92.33±1.09 | 91.32±0.62 | 92.95±0.29 | 71.75±0.25 | 71.68±0.36 |
| DE (EF) | 99.20±0.11 | **93.16±0.70** | 91.76±0.33 | 92.99±0.09 | **72.70±0.39** | 71.67±0.23 |
| SNGP | 98.85±0.22 | 89.50±0.75 | 87.06±1.23 | 91.24±0.46 | 64.68±4.03 | 67.65±1.03 |
| DE (LF) | **99.25±0.00** | 92.33±0.70 | 87.21±0.66 | 92.97±0.13 | 67.05±0.38 | 69.98±0.36 |
| TMC | 98.10±0.14 | 91.17±0.46 | 91.18±1.72 | 91.63±0.28 | 67.68±0.27 | 65.17±0.87 |
| MGP (Ours) | 98.60±0.14 | 92.33±0.70 | **92.06±0.96** | **93.00±0.33** | 70.00±0.53 | **72.30±0.19** |

Table 2: In-domain test ECE ↓

| | Dataset | | | | | |
|---|---|---|---|---|---|---|
| Method | Handwritten | CUB | PIE | Caltech101 | Scene15 | HMDB |
| MC Dropout | 0.009±0.000 | 0.069±0.017 | 0.299±0.005 | 0.017±0.003 | 0.181±0.003 | 0.388±0.004 |
| DE (EF) | 0.007±0.000 | 0.054±0.010 | 0.269±0.004 | 0.036±0.001 | 0.089±0.003 | 0.095±0.003 |
| SNGP | 0.023±0.004 | 0.200±0.010 | 0.852±0.012 | 0.442±0.004 | 0.111±0.063 | 0.227±0.010 |
| DE (LF) | 0.292±0.001 | 0.270±0.009 | 0.567±0.006 | 0.023±0.002 | 0.319±0.005 | 0.270±0.003 |
| TMC | 0.013±0.002 | 0.141±0.002 | **0.072±0.011** | 0.068±0.002 | 0.180±0.004 | 0.594±0.008 |
| MGP (Ours) | **0.006±0.004** | **0.038±0.007** | 0.079±0.007 | **0.009±0.003** | **0.062±0.006** | **0.036±0.003** |

into single feature. The selected single-view baselines are as follows: **MC Dropout** [9] with dropout rate of 0.2, **Deep Ensemble (DE)** [23] with 5 models, and **SNGP**'s GP layer [29]. The multi-view baselines are following: **Deep Ensemble (DE)** with late fusion (LF) technique [2] where one model was trained for one view and predictions of all views were combined as single prediction by averaging the predictions, and **TMC** [11]. For details of model settings, see Appendix.

**Results** For every metric, mean and standard deviation from five runs with different random seeds are reported. The results of multi-view classifiers are generated from the combined predictions. In Table 1 and 2, we provide test accuracy and test expected calibration error (ECE) [10] without adding noise (the definition of ECE is provided in Appendix). In terms of test accuracy, MGP outperforms DE(LF) over 4 out of 6 datasets and TMC over all the datasets. Also, the test ECE of MGP outperforms both DE(EF) and DE(LF) over all the datasets and TMC over 5 out of 6 datasets. Figure 3 shows the test accuracy with respect to the standard deviation of Gaussian noise, representing the domain-shift test accuracy. This highlights that MGP is robust to noise while the accuracy of others degrades significantly. We also report the average accuracy in Table 3.

Table 3: Average test accuracy with Gaussian noise (std from 0.01 to 10) added to half of the views.

| Method | Dataset | | | | | |
| | Handwritten | CUB | PIE | Caltech101 | Scene15 | HMDB |
|---|---|---|---|---|---|---|
| MC Dropout | 82.15±0.17 | 76.08±0.61 | 64.65±0.77 | 73.45±0.11 | 48.97±0.33 | 42.63±0.08 |
| DE (EF) | 82.16±0.18 | 76.94±0.82 | 65.53±0.20 | 73.99±0.19 | 49.45±0.35 | 41.92±0.06 |
| SNGP | 72.46±0.41 | 61.27±1.24 | 56.52±0.69 | 56.57±0.17 | 38.19±1.86 | 37.49±0.42 |
| DE (LF) | 95.63±0.08 | 76.16±0.28 | 67.69±0.35 | 81.85±0.14 | 50.13±0.27 | 43.01±0.19 |
| TMC | 82.44±0.15 | 74.19±0.69 | 62.18±0.80 | 71.77±0.22 | 42.52±0.29 | 36.61±0.30 |
| MGP (Ours) | **97.66±0.12** | **85.48±0.25** | **90.97±0.19** | **92.68±0.23** | **65.74±0.56** | **67.02±0.21** |

Table 4: Out-of-domain detection results with CIFAR10-C

| Method | Test accuracy ↑ | ECE ↓ | OOD AUROC ↑ | |
| | | | SVHN | CIFAR100 |
|---|---|---|---|---|
| MC Dropout | 74.76±0.27 | **0.013±0.002** | 0.788±0.022 | 0.725±0.014 |
| DE (EF) | 72.95±0.13 | 0.154±0.048 | 0.769±0.008 | 0.721±0.014 |
| SNGP | 61.51±0.30 | 0.020±0.003 | 0.753±0.026 | 0.705±0.024 |
| DE (LF) | **75.40±0.06** | 0.095±0.001 | 0.722±0.016 | 0.693±0.006 |
| TMC | 72.42±0.05 | 0.108±0.001 | 0.681±0.004 | 0.675±0.006 |
| MGP (Ours) | 73.30±0.05 | 0.018±0.001 | **0.803±0.007** | **0.748±0.007** |

## 4.3 OOD Samples Detection

**Experimental Settings** Similar to experimental settings in [29], we investigated OOD detection test with CIFAR10-C [13], SVHN [36], and CIFAR100 [22]. We used CIFAR10-C as a multi-view dataset which is a corrupted version of CIFAR10 with 15 different types of corruption and 5 severity levels. The first three corruption types were selected as three multi-view inputs with severity levels of 1, 3, and 5 respectively in order to have variety of noise levels. Each view was trained with CIFAR10-C and tested with SVHN and CIFAR100 as OOD detection tests. Detecting SVHN samples is an easier task since SVHN is distinct from CIFAR10-C, and detecting CIFAR100 samples is a more difficult task since CIFAR100 looks similar to CIFAR10-C. For all methods, we used the Inception v3 [41] pre-trained with ImageNet as the feature extractor without further training it. Details of experimental settings are reported in Appendix.

**Results** Table 4 shows in-domain test accuracy, ECE for CIFAR10-C, and OOD results with SVHN and CIFAR100. MGP's OOD results significantly outperform the others with comparable test accuracy and ECE. Especially, MGP outperforms TMC over all the metrics. Note that the difference in ECE and OOD AUROC between TMC and MGP is considerably larger than the difference in test accuracy. The similar pattern is observed when DE(LF) and MGP are compared, where test accuracy of DE(LF) is slightly higher than MGP, but MGP outperforms DE(LF) with ECE and OOD AUROC. This highlights that although multi-view baselines can provide accurate predictions with in-domain samples, their calibration and uncertainty estimation could be limited, which aligns with [37, 34, 29, 45]. To validate that the predictive uncertainty of MGP could identify OOD samples, we also provide the predictive uncertainty with both testing sets of SVHN and CIFAR100 in Figure 4.

## 5 Conclusion

In this work, we have proposed a new multi-view classification framework for better uncertainty estimation and out-of-domain sample detection, where we associate each view with an uncertainty-aware GP classifier and combine the predictions of all the views with the PoE mechanism. With both the synthetic and the real-world data, we empirically demonstrated that our method is robust to domain-shift and aware of OOD samples, outperforming other baselines in ECE and OOD detection scores. Our method is not limited to a particular feature extractor and can be attached on top of existing feature extractors. A possible limitation of our work is that the weight term that balances predictive distributions is introduced in a sub-optimal way. We leave optimizing it as future work.

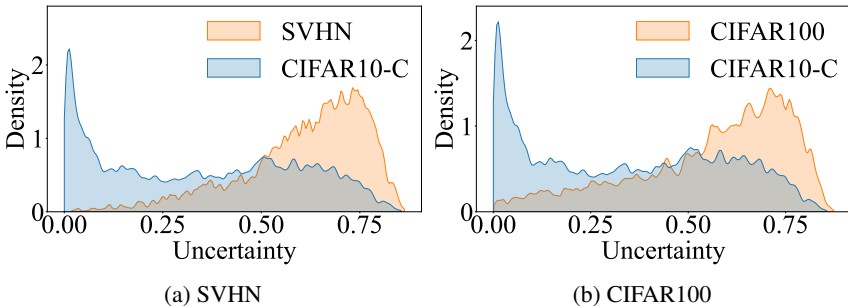

Figure 4: Uncertatinty density of MGP with OOD testing sets: (a) SVHN and (b) CIFAR100.

## Acknowledgments and Disclosure of Funding

**Acknowledgments** We would like to thank reviewers for their time and effort to review this paper. We sincerely appreciate the comments that enhanced and extended our work.

**Funding** This work was part of the Predicting fracture outcomes from clinical registry data using artificial intelligence supplemented models for evidence-informed treatment (PRAISE) study. This study is funded by the National Health and Medical Research Council of Australia Ideas Grant (NHMRC- APP2003537).

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
