# Uncertainty Estimation for Multi-view Data: The Power of Seeing the Whole Picture
## Appendix

## A Proofs and Derivations

**Proof of Lemma 1**

*Proof.* With $x_{2:n} = [x_2, \cdots, x_n]$, we rewrite the KL divergence between $p(x)$ and $q(x)$ as follows:

$$\text{KL}\left[p(x)||q(x)\right] = \int_{x_1 \in \mathcal{X}_1} \int_{x_{2:n} \in \mathcal{X}_{2:n}} p_1(x_1)p_{2:n}(x_{2:n}) \log \left( \frac{p_1(x_1)p_{2:n}(x_{2:n})}{q_1(x_1)q_{2:n}(x_{2:n})} \right) dx_{2:n}dx_1 \quad (1)$$

$$= \int_{x_1 \in \mathcal{X}_1} \int_{x_{2:n} \in \mathcal{X}_{2:n}} p_1(x_1)p_{2:n}(x_{2:n}) \log \left( \frac{p_1(x_1)}{q_1(x_1)} \right) dx_{2:n}dx_1 +$$

$$\int_{x_1 \in \mathcal{X}_1} \int_{x_{2:n} \in \mathcal{X}_{2:n}} p_1(x_1)p_{2:n}(x_{2:n}) \log \left( \frac{p_{2:n}(x_{2:n})}{q_{2:n}(x_{2:n})} \right) dx_{2:n}dx_1 \quad (2)$$

$$= \underbrace{\int_{x_{2:n} \in \mathcal{X}_{2:n}} p_{2:n}(x_{2:n}) dx_{2:n}}_{=1} \int_{x_1 \in \mathcal{X}_1} p_1(x_1) \log \left( \frac{p_1(x_1)}{q_1(x_1)} \right) dx_1$$

$$+ \underbrace{\int_{x_1 \in \mathcal{X}_1} p_1(x_1) dx_1}_{=1} \int_{x_{2:n} \in \mathcal{X}_{2:n}} p_{2:n}(x_{2:n}) \log \left( \frac{p_{2:n}(x_{2:n})}{q_{2:n}(x_{2:n})} \right) dx_{2:n} \quad (3)$$

$$= \text{KL}\left[p(x_1)||q(x_1)\right] + \text{KL}\left[p_{2:n}(x_{2:n})||q_{2:n}(x_{2:n})\right] \quad (4)$$

We can repeat Equation (1)-(3) on the second term of Equation (4) to prove the lemma. □

**Proof of Theorem 2** We let the true posterior $p(\boldsymbol{f}|\boldsymbol{X}, \widetilde{\boldsymbol{y}}_c)$ factorized as $p(\boldsymbol{f}|\boldsymbol{X}, \widetilde{\boldsymbol{y}}_c) = \prod_v p(\boldsymbol{f}_v|\widetilde{\boldsymbol{y}}_c)$. By using the Lemma 1, we form the KL between $[q_{PoE}(\boldsymbol{f})$ and $p(\boldsymbol{f}|\boldsymbol{X}, \widetilde{\boldsymbol{y}}_c)$ as:

$$\text{KL}\left[q_{PoE}(\boldsymbol{f})||p(\boldsymbol{f}|\boldsymbol{X}, \widetilde{\boldsymbol{y}}_c)\right] = \text{KL}\left[\prod_v q(\boldsymbol{f}_v)|| \prod_v p(\boldsymbol{f}_v|\widetilde{\boldsymbol{y}}_c)\right] \quad (5)$$

$$= \sum_v \text{KL}\left[q(\boldsymbol{f}_v)||p(\boldsymbol{f}_v|\widetilde{\boldsymbol{y}}_c)\right] \quad (6)$$

**Derivation of ELBO** By expanding Equation (6), we formulate ELBOs for each view as follows:

$$\sum_v \text{KL}\left[q(\boldsymbol{f}_v)||p(\boldsymbol{f}_v|\widetilde{\boldsymbol{y}}_c)\right]$$

$$= \sum_v \int q(\boldsymbol{f}_v) \log \frac{q(\boldsymbol{f}_v)}{p(\boldsymbol{f}_v|\widetilde{\boldsymbol{y}}_c)} \, d\boldsymbol{f}_v$$

$$= \sum_v \left[ \int q(\boldsymbol{f}_v) \log q(\boldsymbol{f}_v) \, d\boldsymbol{f}_v - \int q(\boldsymbol{f}_v) \log \frac{p(\widetilde{\boldsymbol{y}}_c|\boldsymbol{f}_v)p(\boldsymbol{f}_v)}{p(\widetilde{\boldsymbol{y}}_c)} \, d\boldsymbol{f}_v \right]$$

$$= \sum_v \left[ \int q(\boldsymbol{f}_v) \log \frac{q(\boldsymbol{f}_v)}{p(\boldsymbol{f}_v)} \, d\boldsymbol{f}_v - \int q(\boldsymbol{f}_v) \log p(\widetilde{\boldsymbol{y}}_c|\boldsymbol{f}_v) \, d\boldsymbol{f}_v + \log p(\widetilde{\boldsymbol{y}}_c) \right] \quad (7)$$

By rearranging Equation (7), we obtain:

$$\log p(\widetilde{\boldsymbol{y}}_c) \geq \sum_v \left[ \int q(\boldsymbol{f}_v) \log p(\widetilde{\boldsymbol{y}}_c|\boldsymbol{f}_v)\, d\boldsymbol{f}_v - \int q(\boldsymbol{f}_v) \log \frac{q(\boldsymbol{f}_v)}{p(\boldsymbol{f}_v)}\, d\boldsymbol{f}_v \right]$$

$$= \sum_v \left[ \int q(\boldsymbol{f}_v) \log p(\widetilde{\boldsymbol{y}}_c|\boldsymbol{f}_v)\, d\boldsymbol{f}_v - \iint q(\boldsymbol{f}_v, \boldsymbol{u}_v) \log \frac{q(\boldsymbol{f}_v)}{p(\boldsymbol{f}_v)}\, d\boldsymbol{u}_v\, d\boldsymbol{f}_v \right]$$

$$= \sum_v \left[ \int q(\boldsymbol{f}_v) \log p(\widetilde{\boldsymbol{y}}_c|\boldsymbol{f}_v)\, d\boldsymbol{f}_v - \iint q(\boldsymbol{f}_v, \boldsymbol{u}_v) \log \frac{q(\boldsymbol{f}_v, \boldsymbol{u}_v)/p(\boldsymbol{u}_v|\boldsymbol{f}_v)}{p(\boldsymbol{f}_v, \boldsymbol{u}_v)/p(\boldsymbol{u}_v|\boldsymbol{f}_v)}\, d\boldsymbol{u}_v\, d\boldsymbol{f}_v \right]$$

$$= \sum_v \left[ \int q(\boldsymbol{f}_v) \log p(\widetilde{\boldsymbol{y}}_c|\boldsymbol{f}_v)\, d\boldsymbol{f}_v - \iint q(\boldsymbol{f}_v, \boldsymbol{u}_v) \log \frac{q(\boldsymbol{f}_v, \boldsymbol{u}_v)}{p(\boldsymbol{f}_v, \boldsymbol{u}_v)}\, d\boldsymbol{u}_v\, d\boldsymbol{f}_v \right]$$

$$= \sum_v \left[ \int q(\boldsymbol{f}_v) \log p(\widetilde{\boldsymbol{y}}_c|\boldsymbol{f}_v)\, d\boldsymbol{f}_v - \iint q(\boldsymbol{f}_v, \boldsymbol{u}_v) \log \frac{p(\boldsymbol{f}_v|\boldsymbol{u}_v)q(\boldsymbol{u}_v)}{p(\boldsymbol{f}_v|\boldsymbol{u}_v)p(\boldsymbol{u}_v)}\, d\boldsymbol{u}_v\, d\boldsymbol{f}_v \right]$$

$$= \sum_v \left[ \int q(\boldsymbol{f}_v) \log p(\widetilde{\boldsymbol{y}}_c|\boldsymbol{f}_v)\, d\boldsymbol{f}_v - \iint q(\boldsymbol{f}_v, \boldsymbol{u}_v) \log \frac{q(\boldsymbol{u}_v)}{p(\boldsymbol{u}_v)}\, d\boldsymbol{u}_v\, d\boldsymbol{f}_v \right]$$

$$= \sum_v \left[ \int q(\boldsymbol{f}_v) \log p(\widetilde{\boldsymbol{y}}_c|\boldsymbol{f}_v)\, d\boldsymbol{f}_v - \mathrm{KL}\left[q(\boldsymbol{u}_v)||p(\boldsymbol{u}_v)\right] \right] = \sum_v \mathrm{ELBO}_v \qquad (8)$$

The KL term in Equation (8) has an analytical expression because both $q(\boldsymbol{u}_v)$ and $p(\boldsymbol{u}_v)$ are Gaussian distributions. However, the log likelihood term is not analytical yet. We further factorize the likelihood across data points as:

$$\int q(\boldsymbol{f}_v) \log p(\widetilde{\boldsymbol{y}}_c|\boldsymbol{f}_v)\, d\boldsymbol{f}_v = \int q(\boldsymbol{f}_v) \log \left( \prod_{i=1}^N p(\widetilde{y}_{i,c}|\boldsymbol{f}_{i,v}) \right) d\boldsymbol{f}_v$$

$$= \int q(\boldsymbol{f}_v) \sum_{i=1}^N \log p(\widetilde{y}_{i,c}|\boldsymbol{f}_{i,v})\, d\boldsymbol{f}_v$$

$$= \sum_{i=1}^N \left( \int q(\boldsymbol{f}_v) \log p(\widetilde{y}_{i,c}|\boldsymbol{f}_{i,v})\, d\boldsymbol{f}_v \right)$$

$$= \sum_{i=1}^N \left( \iint q(\boldsymbol{f}_{\boldsymbol{j},v}, \boldsymbol{f}_{i,v}) \log p(\widetilde{y}_{i,c}|\boldsymbol{f}_{i,v})\, d\boldsymbol{f}_{\boldsymbol{j}}\, d\boldsymbol{f}_v \right), \;\; \boldsymbol{j} = \{n\}_{n=1}^N \setminus \{i\}$$

$$= \sum_{i=1}^N \left( \int q(\boldsymbol{f}_{i,v}) \log p(\widetilde{y}_{i,c}|\boldsymbol{f}_{i,v})\, d\boldsymbol{f}_{i,v} \right)$$

$$= \sum_{i=1}^N \mathbb{E}_{q(\boldsymbol{f}_{i,v})}\left[\log p(\widetilde{y}_{i,c}|\boldsymbol{f}_{i,v})\right] \qquad (9)$$

By substituting Equation (9) in Equation (8) and introducing $\beta$ to control the regularization term, the ELBO for a view is defined as:

$$\mathrm{ELBO}_v = \sum_{i=1}^N \mathbb{E}_{q(\boldsymbol{f}_{v,i})}\left[\log p(\widetilde{y}_{i,c}|\boldsymbol{f}_{v,i})\right] - \beta \cdot \mathrm{KL}\left[q(\boldsymbol{u}_v)||p(\boldsymbol{u}_v)\right] \qquad (10)$$

Note that $q(\boldsymbol{f}_v)$ has an analytical solution because the conditional prior $p(\boldsymbol{f}_v|\boldsymbol{u}_v) = \mathcal{N}(\boldsymbol{f}_v; \boldsymbol{K}_{NM}\boldsymbol{K}_{MM}^{-1}\boldsymbol{u}_v, \boldsymbol{K}_{NN} - \boldsymbol{K}_{NM}\boldsymbol{K}_{MM}^{-1}\boldsymbol{K}_{NM}^{\mathrm{T}})$ and the marginal variational distribution $q(\boldsymbol{u}_v) = \mathcal{N}(\boldsymbol{u}_v; \boldsymbol{m}_v \boldsymbol{S}_v)$ are both Gaussian distributions. By using Gaussian linear transforma-

tion and integrating $\boldsymbol{u}_v$ out, we can derive the solution as follows:

$$
\begin{aligned}
q(\boldsymbol{f}_v) &:= \int p(\boldsymbol{f}_v | \boldsymbol{u}_v) q(\boldsymbol{u}_v) \, d\boldsymbol{u}_v \\
&= \int \mathcal{N}(\boldsymbol{f}_v; \boldsymbol{K}_{NM} \boldsymbol{K}_{MM}^{-1} \boldsymbol{u}_v, \boldsymbol{K}_{NN} - \boldsymbol{K}_{NM} \boldsymbol{K}_{MM}^{-1} \boldsymbol{K}_{NM}^{\mathrm{T}}) \mathcal{N}(\boldsymbol{u}_v; \boldsymbol{m}_v \boldsymbol{S}_v) \, d\boldsymbol{u}_v \\
&= \int \mathcal{N}(\boldsymbol{f}_v; \boldsymbol{K}_{NM} \boldsymbol{K}_{MM}^{-1} \boldsymbol{m}_v, \boldsymbol{K}_{NM} \boldsymbol{K}_{MM}^{-1} \boldsymbol{S}_v (\boldsymbol{K}_{NM} \boldsymbol{K}_{MM}^{-1})^{\mathrm{T}} + \boldsymbol{K}_{NN} \\
&\qquad\qquad - \boldsymbol{K}_{NM} \boldsymbol{K}_{MM}^{-1} \boldsymbol{K}_{NM}^{\mathrm{T}}) \mathcal{N}(\boldsymbol{u}_v; \boldsymbol{m}_v \boldsymbol{S}_v) \, d\boldsymbol{u}_v \\
&= \mathcal{N}(\boldsymbol{f}_v; \boldsymbol{K}_{NM} \boldsymbol{K}_{MM}^{-1} \boldsymbol{m}_v, \boldsymbol{K}_{NM} \boldsymbol{K}_{MM}^{-1} \boldsymbol{S}_v (\boldsymbol{K}_{NM} \boldsymbol{K}_{MM}^{-1})^{\mathrm{T}} + \boldsymbol{K}_{NN} \\
&\qquad\qquad - \boldsymbol{K}_{NM} \boldsymbol{K}_{MM}^{-1} \boldsymbol{K}_{NM}^{\mathrm{T}}) \int \mathcal{N}(\boldsymbol{u}_v; \boldsymbol{m}_v \boldsymbol{S}_v) \, d\boldsymbol{u}_v \\
&= \mathcal{N}(\boldsymbol{f}_v; \boldsymbol{K}_{NM} \boldsymbol{K}_{MM}^{-1} \boldsymbol{m}_v, \boldsymbol{K}_{NM} \boldsymbol{K}_{MM}^{-1} \boldsymbol{S}_v (\boldsymbol{K}_{NM} \boldsymbol{K}_{MM}^{-1})^{\mathrm{T}} + \boldsymbol{K}_{NN} - \boldsymbol{K}_{NM} \boldsymbol{K}_{MM}^{-1} \boldsymbol{K}_{NM}^{\mathrm{T}})
\end{aligned}
$$

**Inference**  Given test samples $\boldsymbol{X}_* = \{\boldsymbol{X}_{v,*}\}_{v=1}^{V}$, the predictive distribution $p(\boldsymbol{f}_{*,v} | \widetilde{\boldsymbol{y}}_c)$ of single view is estimated by the variational distribution as:

$$
\begin{aligned}
p(\boldsymbol{f}_{*,v} | \widetilde{\boldsymbol{y}}_c) &= \iint p(\boldsymbol{f}_{*,v} | \boldsymbol{f}, \boldsymbol{u}_v | \widetilde{\boldsymbol{y}}_c) p(\boldsymbol{f}, \boldsymbol{u}_v | \boldsymbol{u}_v) \, d\boldsymbol{f} \, d\boldsymbol{u}_v \\
&\approx \iint p(\boldsymbol{f}_{*,v} | \boldsymbol{f}, \boldsymbol{u}_v) q(\boldsymbol{f}, \boldsymbol{u}_v) \, d\boldsymbol{f} \, d\boldsymbol{u}_v \\
&= \iint p(\boldsymbol{f}_{*,v} | \boldsymbol{f}, \boldsymbol{u}_v) p(\boldsymbol{f} | \boldsymbol{u}_v) q(\boldsymbol{u}_v) \, d\boldsymbol{f} \, d\boldsymbol{u}_v \\
&= \int p(\boldsymbol{f}_{*,v} | \boldsymbol{u}_v) q(\boldsymbol{u}_v) \, d\boldsymbol{u}_v \\
&= q(\boldsymbol{f}_{*,v}) \quad\quad\quad\quad\quad\quad\quad\quad\quad\quad\quad\quad\quad\quad\quad\quad\quad\quad (11)
\end{aligned}
$$

where $p(\boldsymbol{f}_{*,v} | \boldsymbol{u}_v)$ can be formed by the joint prior distribution of:

$$
\begin{bmatrix} \boldsymbol{f}_{*,v} \\ \boldsymbol{u}_v \end{bmatrix} \sim \mathcal{N}\left(\boldsymbol{0}, \begin{bmatrix} \boldsymbol{K}_{**} & \boldsymbol{K}_{*M} \\ \boldsymbol{K}_{*M}^{\mathrm{T}} & \boldsymbol{K}_{MM} \end{bmatrix}\right)
$$

# B    Detailed Experimental Settings and Results

In this section, we provide detailed experimental settings and additional experimental results for the synthetic dataset experiment in Appendix B.1, the robustness to noise experiment in Appendix B.2, and OOD samples detection experiment in Appendix B.3.

## B.1    Synthetic Dataset Experiment

**Dataset**  The original moon dataset in Scikit-learn[1] has two sets of 2D data points: upper unit circle points (class 1) and lower unit circle points (class 2). We modified the original code by changing the radius of circle with three radius values (view 1: 1.7, view 2: 1.0, and view 3: 0.3) with a fixed random state. We generated 1,000 points for each set with a different radius, forming an individual input view. The third view with radius 0.3 was further translated to make the points overlapping, representing a noisy view. OOD samples were generated by randomly sampling 200 points from a normal distribution with standard deviation of 0.04.

**Feature Extractor**  To make the comparisons between methods be fair, we used the same feature extractor architecture. Similar to [8], we used a non-trainable fully connected layer to project the input to a hidden dimension. Then, six residual fully connected layers with the same hidden dimension are stacked. We used 128 units as the hidden dimension.

---

[1]`https://scikit-learn.org/stable/modules/generated/sklearn.datasets.make_moons.html`

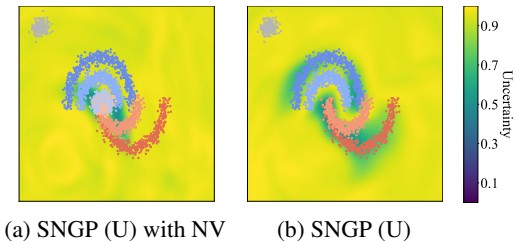

(a) SNGP (U) with NV    (b) SNGP (U)

Figure 1: Uncertainty surfaces (U) of SNGP (a) with the noisy view (NV) and (b) without the noisy view.

**Implementation Details**    On top of the feature extractor, we used different output layers as follows:

- **SNGP**'s GP layer: We followed the settings provided by its authors in their tutorial[2] with `gp_cov_momentum = -1` for computing the model's covariance and $\lambda = \pi/8$ for the mean-filed estimation. Since SNGP is a unimodal model, we used early fusion method to concatenate input points into $\boldsymbol{X} \in \mathbb{R}^{N \times 2V}$. The constant learning rate of 0.001 was used.

- **DE(LF)**: For each view, a fully connected layer was trained individually with the constant learning rate of 0.001. Given $i^{th}$ test sample, predictive probabilities from all views were averaged.

- **TMC**: We used the identical architecture proposed by [4] where each view is built with a fully connected layer with $l^2$ regularization coefficient of 0.0001. `lambda_epochs = 10` for annealing KL term was used.

- **MGP (Ours)**: GPs with $\alpha_\epsilon = 0.001$ and 200 inducing points were used for each view. $l_v$ and $\sigma_v^2$ were initialized with $\{1.0\}_{d=1}^D$ where $D$ is the output dimension of the feature extractor, and $Z_v$ was initialized with the first 200 samples of the training set. To stabilize the training, we first trained GP layer for 10 warm-up epochs without training the feature extractor with the learning rate of 0.01 which was linearly decreased to 0.003. Then, the entire model was trained with the learning rate of 0.003. We used $\beta = 1$ for the regularization coefficient. Our framework is based on GPflow [9].

We implemented all the methods in Tensorflow and trained them for 30 epochs with the Adam optimizer [6] on single Nvidia GeForce RTX 3090 (24GB) GPU.

**Uncertainty Quantification**    TMC and SNGP quantify predictive uncertainty based on the Dempster–Shafer theory [1]. TMC's output has explicit expression of uncertainty quantity, and SNGP estimates the uncertainty with output logits as:

$$\boldsymbol{U}(\boldsymbol{x}_i) = \frac{C}{C + \sum_{c=1}^{C} \exp\left(h_c(\boldsymbol{x}_i)\right)} \tag{12}$$

where $h_c(\boldsymbol{x}_i)$ is the $c^{th}$ class logit of SNGP. The difference between uncertainty surfaces of SNGP trained with the noisy view and without the noisy view is shown in Figure 1.

For MGP, we used the sum of predictive variance over all classes as uncertainty, which is another way of quantifying uncertainty [2]. The uncertainty surfaces of each view are shown in Figure 2.

**SNGP with Spectral Normalization**    The original SNGP uses the residual feature extractor with spectral normalization. We implemented the same experiment with the feature extractor using spectral normalization with `norm_multiplier = 0.9` as introduced in the tutorial. The results are plotted in Figure 3.

### B.2    Robustness to Noise Experiment

**Implementation Details**    We used the same datasets of the TMC's datasets (Handwritten, CUB, PIE, Caltech101, Scene15, and HMDB). For details of the datasets, refer to [4]. The experimental

---

[2]`https://www.tensorflow.org/tutorials/understanding/sngp`

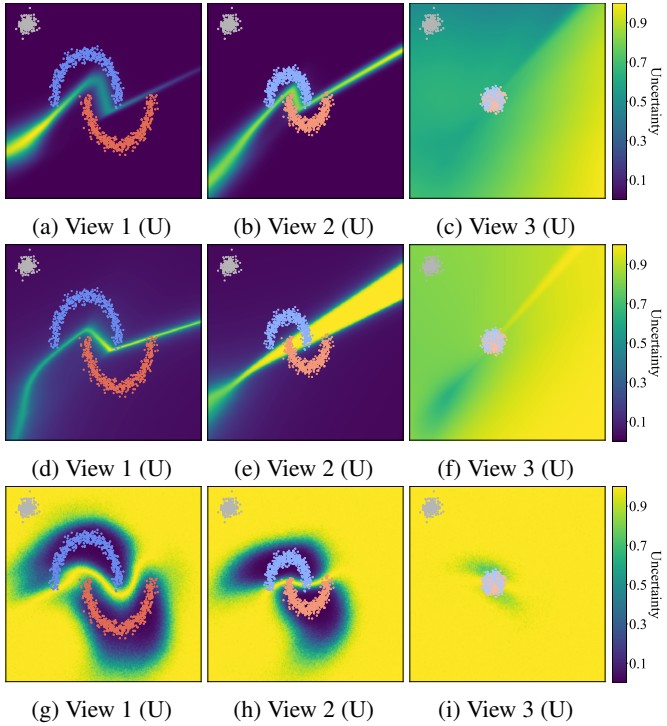

(a) View 1 (U)  (b) View 2 (U)  (c) View 3 (U)

(d) View 1 (U)  (e) View 2 (U)  (f) View 3 (U)

(g) View 1 (U)  (h) View 2 (U)  (i) View 3 (U)

Figure 2: Uncertainty surfaces (U) of view 1, 2, and 3 from DE(LF) (a)-(c), TMC (d)-(f), and MGP (g)-(i).

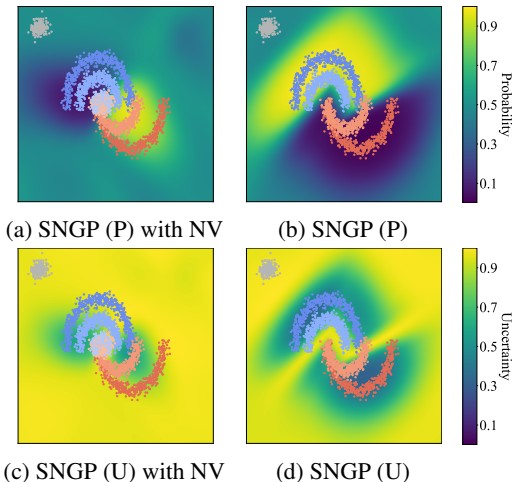

(a) SNGP (P) with NV  (b) SNGP (P)

(c) SNGP (U) with NV  (d) SNGP (U)

Figure 3: Top row: Predictive probability surfaces (P) of SNGP using spectral normalization with the noisy view (NV) (left) and without the noisy view (right); bottom row: Uncertainty surfaces of SNGP using spectral normalization (U) with the noisy view (NV) (left) and without the noisy view (right).

settings are similar to Appendix B.1 except that the feature extractor was not used because the datasets are feature sets. In addition to the methods used in B.1, we implemented MC Dropout and DE(EF) models with the seetings as follows:

- **MC Dropout**: We used a dropout layer with the dropout rate of 0.2 and a fully connected layer on top of the dropout layer. During inference, 100 samples were used to make a prediction. We used the early fusion method to concatenate multi-view features into unimodal feature.

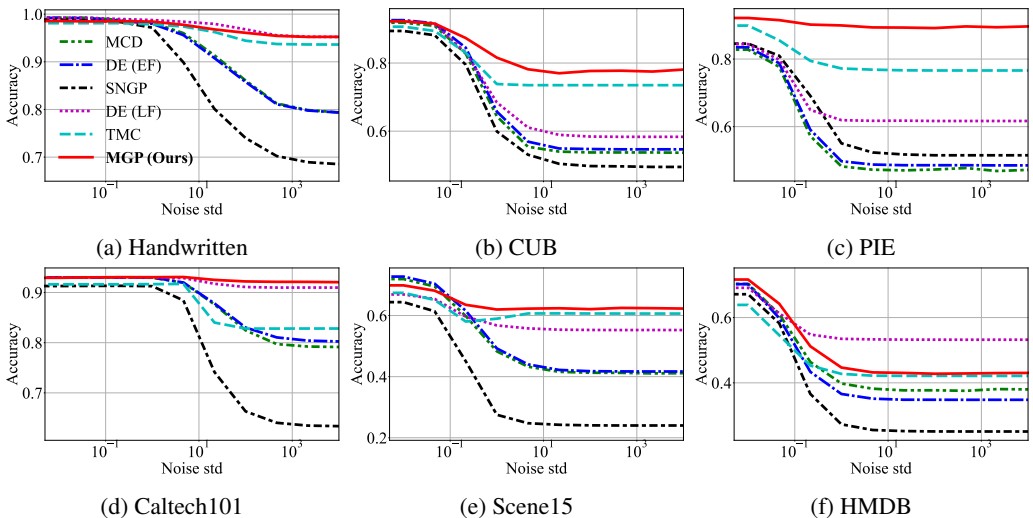

Figure 4: Domain-shift test accuracy where Gaussian noise is added to half of the views.

- **DE (EF)**: We used a fully connected layer for each model. In total, 5 models were trained individually, and their predictions were averaged.

For evaluation, we estimated expected calibration error (ECE) [3] to measure the difference of model's accuracy and confidence by:

$$\text{ECE} = \sum_{k=1}^{K} \frac{|B_k|}{N} \left| acc(B_k) - conf(B_k) \right|$$

where $K$ is the number of bins, and $B_k$ is partitioned predictions of the bins. In our experiments, we set $K = 15$.

**Normalization of Input**  In our experiment, we normalized the datasets first and added noise to half of the views to maintain the same impact of noise on all the views. However, TMC added the noise first and normalized the noisy inputs. The experimental results with this setting are plotted in Figure 4 and Table 1. For each noise level, the average result of all combinations of selecting noisy views (i.e., $\binom{V}{V/2}$ configurations) is reported.

Table 1: Average test accuracy with Gaussian noise (std from 0.01 to 10,000) added to half of the views.

| | Dataset | | | | | |
|---|---|---|---|---|---|---|
| Method | Handwritten | CUB | PIE | Caltech101 | Scene15 | HMDB |
| MC Dropout | 91.69±0.36 | 67.85±0.65 | 58.23±0.40 | 87.73±0.35 | 51.91±0.39 | 46.96±0.52 |
| DE (EF) | 91.50±0.25 | 68.93±0.57 | 59.49±0.52 | 88.13±0.10 | 52.70±0.29 | 44.63±0.68 |
| SNGP | 85.78±1.32 | 64.40±1.34 | 62.45±1.45 | 79.65±0.59 | 37.10±2.49 | 37.10±0.33 |
| DE (LF) | **97.69±0.06** | 70.99±0.71 | 68.04±0.62 | _92.14±0.18_ | 58.90±0.76 | **57.10±0.64** |
| TMC | 96.30±0.50 | 79.01±0.85 | _80.30±2.05_ | 87.74±0.27 | _61.92±0.42_ | 47.68±0.84 |
| MGP (Ours) | _97.19±0.14_ | **82.85±0.96** | **90.15±0.33** | **92.65±0.23** | **64.29±0.72** | _51.11±0.74_ |

## B.3  OOD Samples Detection Experiment

**Implementation Details**  For in-domain tests, the original train and test splits of CIFAR10 [7] with corruptions [5] were used. Two OOD testing sets were generated by randomly selecting 5,000 samples (half of the testing set) from CIFAR10-C and 5,000 samples from SVHN or CIFAR100. Predictive uncertainty was used to detect OOD samples with the performance measured by the area under the receiver operating characteristic (AUROC). We used the Inception v3 [10] pre-trained with

ImageNet as a feature extractor without fine-tuning it. To save computational resources, we stored its features first and used them without further processing them.

## C  Parameter Sensitivity of MGP

There are two model parameters introduced in our framework, namely $\alpha_\epsilon$ for the label transformation and the number of inducing points $M$ for each GP expert. In this section, we provide empirical studies of how these parameters affect the model's performance.

### C.1  Label Transformation on In-domain Accuracy

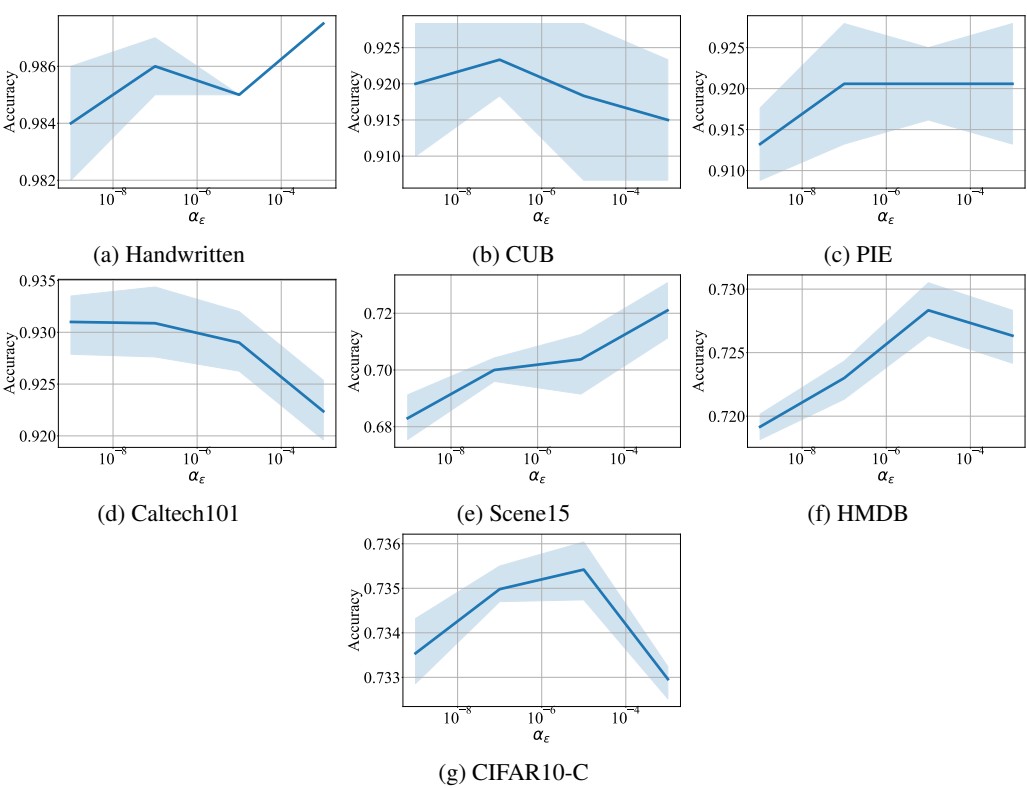

Figure 5: In-domain test accuracy with respect to $\alpha_\epsilon$.

## C.2 Label Transformation on In-domain ECE

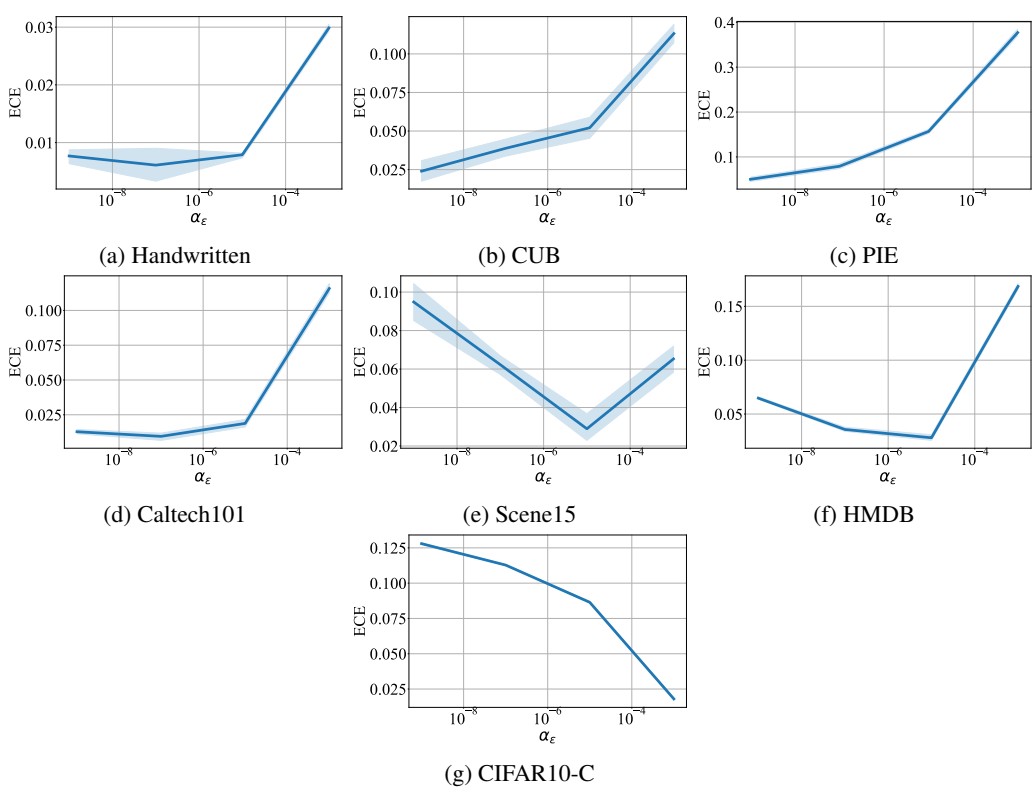

Figure 6: In-domain test ECE with respect to $\alpha_\epsilon$.

## C.3 Label Transformation on OOD AUROC

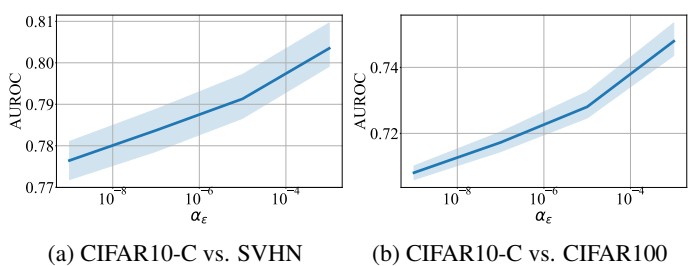

(a) CIFAR10-C vs. SVHN  (b) CIFAR10-C vs. CIFAR100

Figure 7: OOD AUROC with respect to $\alpha_\epsilon$.

## C.4 Number of Inducing Points on Training Time

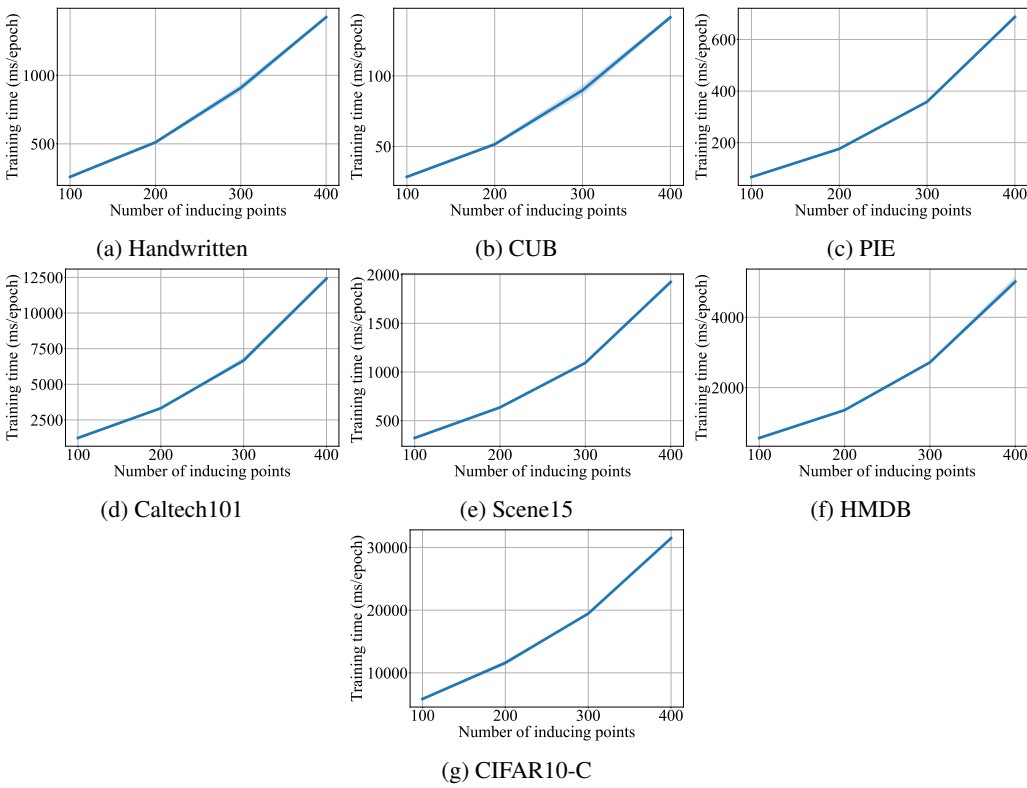

Figure 8: Training time with respect to the number of inducing points.

## C.5   Number of Inducing Points on Testing Time

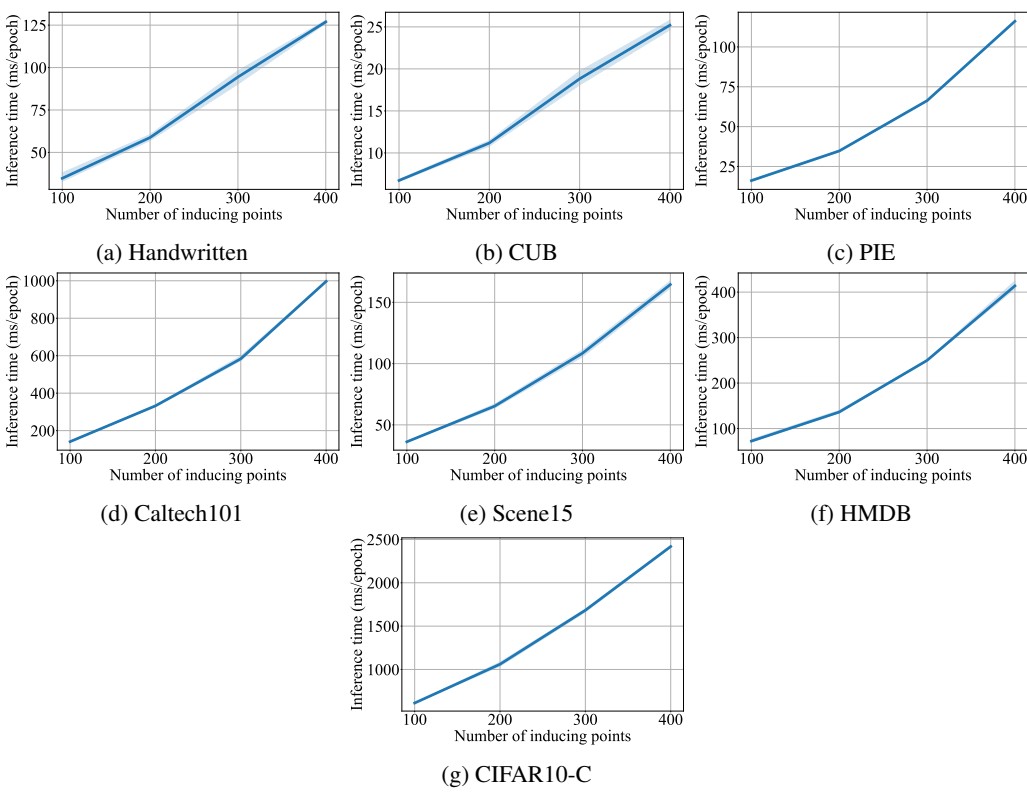

Figure 9: Testing time with respect to the number of inducing points.

## C.6 Number of Inducing Points on In-domain Accuracy

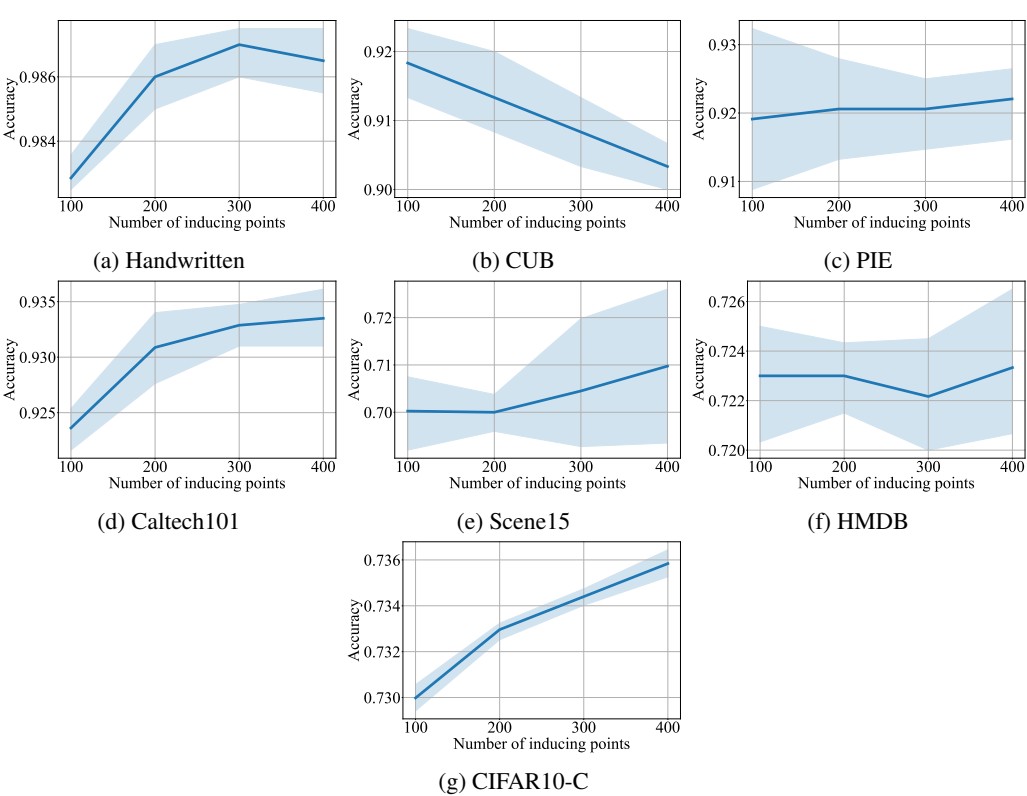

Figure 10: In-domain test accuracy with respect to the number of inducing points.

## C.7 Number of Inducing Points on In-domain ECE

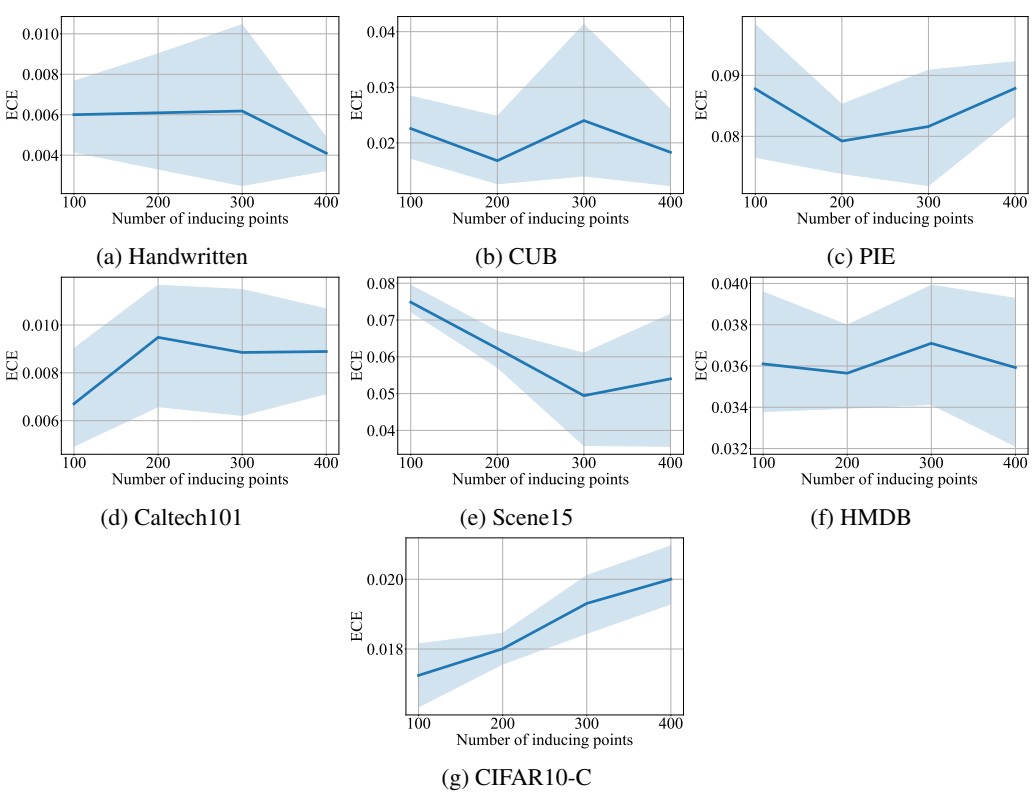

(a) Handwritten

(b) CUB

(c) PIE

(d) Caltech101

(e) Scene15

(f) HMDB

(g) CIFAR10-C

Figure 11: In-domain test ECE with respect to the number of inducing points.

## C.8 Number of Inducing Points on OOD AUROC

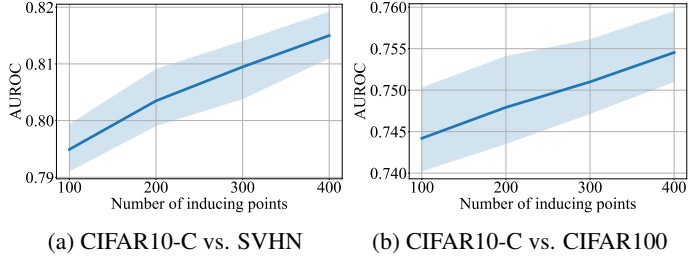

(a) CIFAR10-C vs. SVHN

(b) CIFAR10-C vs. CIFAR100

Figure 12: OOD AUROC with respect to the number of inducing points.

## D  Integral Approximation

In order to compute the posterior of Dirichlet distribution with $\mathbb{E}\left[\pi_{i,c}\right]$ and $\mathbb{V}\left[\pi_{i,c}\right]$, we approximate the integration with the Monte Carlo method. We provide the effect of number of Monte Carlo samples on the in-domain test accuracy in Table 2 and the average inference time in Table 3.

Table 2: The impact of the number of Monte Carlo samples on the in-domain test accuracy.

| # Monte Carlo Samples | Handwritten | CUB | PIE | Caltech101 | Scene15 | HMDB | CIFAR10-C |
|---|---|---|---|---|---|---|---|
| 1 | 98.25±0.40 | 85.50±2.09 | 83.38±2.47 | 91.16±0.40 | 65.55±1.10 | 66.38±0.86 | 70.86±0.36 |
| 10 | 98.50±0.25 | 91.33±2.09 | 90.29±1.90 | 92.86±0.47 | 69.95±1.50 | 71.88±0.72 | 73.01±0.14 |
| 25 | 98.45±0.11 | 92.50±1.02 | 91.62±1.12 | 93.00±0.25 | 69.75±1.16 | 71.95±0.43 | 73.15±0.12 |
| 50 | 98.60±0.14 | 92.33±0.37 | 91.91±0.52 | 93.06±0.33 | 69.80±0.73 | 72.37±0.30 | 73.27±0.12 |
| 75 | 98.50±0.18 | 92.33±1.37 | 91.91±0.52 | 93.08±0.20 | 70.42±0.37 | 72.48±0.07 | 73.20±0.19 |
| 100 | 98.60±0.14 | 92.33±0.70 | 92.06±0.96 | 93.00±0.33 | 70.00±0.53 | 72.30±0.19 | 73.30±0.05 |
| 125 | 98.60±0.14 | 92.50±1.18 | 91.62±0.40 | 93.06±0.31 | 70.18±0.53 | 72.50±0.33 | 73.28±0.04 |

Table 3: The impact of the number of Monte Carlo samples on the average inference time (ms/epoch).

| # Monte Carlo Samples | Handwritten | CUB | PIE | Caltech101 | Scene15 | HMDB | CIFAR10-C |
|---|---|---|---|---|---|---|---|
| 1 | 57.35±2.80 | 7.55±2.69 | 34.87±3.72 | 312.94±8.08 | 66.75±3.20 | 130.67±3.85 | 1049.73±23.77 |
| 10 | 57.60±4.87 | 7.74±6.72 | 35.07±3.82 | 313.34±6.77 | 66.88±3.36 | 130.36±3.70 | 1063.48±26.49 |
| 25 | 58.15±4.81 | 7.70±5.79 | 35.30±3.56 | 324.21±10.58 | 67.51±3.35 | 132.05±3.68 | 1057.31±17.51 |
| 50 | 59.05±6.67 | 7.75±7.04 | 36.31±4.38 | 330.36±11.75 | 68.10±4.97 | 134.06±3.60 | 1059.33±17.86 |
| 75 | 58.96±2.95 | 7.60±2.84 | 36.35±1.24 | 333.41±10.59 | 68.68±5.51 | 136.58±4.56 | 1079.50±24.29 |
| 100 | 58.93±2.70 | 7.71±5.96 | 36.41±3.09 | 333.01±10.34 | 68.84±3.26 | 138.05±4.73 | 1097.04±1.49 |
| 125 | 59.32±2.81 | 7.81±3.11 | 36.99±1.40 | 334.87±7.05 | 69.54±4.90 | 139.99±3.44 | 1111.70±27.36 |

Although we can see improvements in accuracy as the number of Monte Carlo samples increases from 1 to 10 across all the datasets, there is no significant difference when the number of samples becomes large, like 75 vs 100 vs 125. Unsurprisingly, the increase in inference time is observed as the number of Monte Carlo samples increases. However, given the random deviations across epochs, the increasing trend is relatively gradual. In our experiments, we fixed it to 100 samples.

## E  Potential Societal Impacts

As our method is not limited to a specific type of data or model's architecture, it could be applied to various multi-class applications that leverage multi-view data such as vision-language learning, multi-sensor learning, diagnostic classification, scene recognition, and many more. Our method's capability of providing uncertainty estimation may gain trust of multi-view/modal deep learning classifiers from experts in other domains. This would eventually make deep learning models more reliable and trustworthy in real-world settings. As a long-term impact, this work would also raise awareness of transparency of deep learning models.

An unintentional risk of our work is undue trust of the estimated uncertainty. Our uncertainty estimation still has limitations which should be comprehensively studied. Without fully understanding the source of uncertainty, deploying the model to safety-critical applications may result in subsequent risks. We encourage researchers to beware of the model's behaviors in different settings.