# OpenReview forum: "Uncertainty Estimation for Multi-view Data: The Power of Seeing the Whole Picture"
_NeurIPS.cc/2022/Conference — NeurIPS 2022 Accept_

### Official Review · Reviewer_2rPi · 2022-07-02

**Rating:** 6
**Confidence:** 4
**Soundness:** 3 good
**Presentation:** 2 fair
**Contribution:** 3 good

**Summary:**

This paper proposed Multi-view Gaussian Process (MGP) for classification with multi-view and multi-modal data. The MGP modeled each view of data using Gaussian Process (GP) whose classification results are aggregated using product of expert (PoE). Variational inference (VI) is used for model learning (estimation of posterior distribution of latent function $f_v$) and inference (estimation of class probability given test data). Experiment with synthetic data and multiple real-world dataset shows that proposed method achieves comparable in-distribution classification accuracy and better robustness to noise than other state-of-the-art methods that also considers uncertainty in data. The proposed method also showed better performance in out-of-distribution detection (trained with CIFAR10-C and detect samples from SVHN and CIFAR100).

**Questions:**

1. How is the $M$ value determined in Eq. (1) and subsequent experiments?
2. In algorithm 1 step 3, how is $q(f_v)$ computed by Eq. (7)? Is there an analytical solution to the integration?
3. In OOD experiment, how is the three views used for training selected? Can we expect similar results if we attempt training and evaluation with a different choice of views such as in a manner of k-fold cross validation?
4. Can the authors provide some discussion on the computation cost vs. accuracy given Monte Carlo estimation is used for Eq. (14) during inference?
5. In Figure 3, the performance of TMC appears to be comparable to DE (EF) but worse than DE (LF), which is different from what reported in [10], which also had a much wider range of noise magnitude. Can the authors comment on what may contribute to this difference?
6. Typo: Line 69 'applyinh' should be 'applying'

**Limitations:**

The author provided a discussion on societal impacts in the supplementary materials. I have no concern on the limitations and potential negative societal impact of the work.

**Strengths And Weaknesses:**

## Strengths

1. The paper extended GP to model multi-view data and utilize PoE for combining results of different views. The overall framework seems to be novel. The adoption of VI provides a principled way to solve the learning and inference problems. Though this is not groundbreaking as it has been adopted for GP in the past such as [14].
2. Empirical evaluation on multiple real-world datasets show that the proposed method is much more robust than other uncertainty-aware methods in robustness to noise and out-of-distribution data.

## Weakness

1. There is no discussion on how uncertainty is actually computed in MGP. In addition, although the authors did provide illustration in Figure 2 on uncertainty values, there is no equivalent illustration or analysis on real datasets.
2. Although the proposed method to improve the robustness and ability to detect OOD samples of classification model. It comes with a cost of reduced in-domain classification accuracy (at least not as competitive as some other state-of-the-art e.g. DE). It is unclear whether this is a limitation of the classification model (or representation power) or the limitation of the proposed inference framework.
3. The metrics Expected Calibration Error (ECE) is introduced without definition or citation. It is preferred to at least describe the intuition and add citation to the metrics.

---

> ### Author Response · Authors · 2022-08-02
> **Response (1/3)**
>
> We thank the reviewer for the comprehensive review and constructive feedback. We would like to address the reviewer’s comments and questions below:
>
> > 1. “There is no discussion on how uncertainty is actually computed in MGP. In addition, although the authors did provide illustration in Figure 2 on uncertainty values, there is no equivalent illustration or analysis on real datasets.”
>
> Thank you for pointing this out. The uncertainty is computed by the sum of predictive variance in Equation (14) over the label classes. It is stated in [Supplementary, L59], but we clarified this in the main text based on the comment [L203]. Also, we added Figure 5 in our supplementary material which shows the predictive uncertainty with CIFAR10-C which is an in-domain dataset and SVHN and CIFAR100 which are OOD datasets. We would like to highlight from Figure 5 (Appendix) that the uncertainty of MGP could identify the OOD samples similar to the illustrations in Figure 1 and 2 in the main text.
>
> > 2. “It is unclear whether this is a limitation of the classification model (or representation power) or the limitation of the proposed inference framework.”
>
> There is a known trade-off between in-domain accuracy and the generalizability for better uncertainty estimation [R1]. Deep ensembles have shown higher accuracy with in-domain testing samples but often compromise calibration and OOD detection performance shown in other studies as well such as [29,43,R2,R3]. In addition, deep ensembles require multiple ensemble members to be trained, which requires higher computational and memory costs. Therefore, our primary focus, similar to [29,43,R2,R3], is to maintain in-domain accuracy comparable to other baselines while improving the robustness of noise and OOD detection performance.
>
> > 3. “The metrics Expected Calibration Error (ECE) is introduced without definition or citation. It is preferred to at least describe the intuition and add citation to the metrics.”
>
> We thank the reviewer for the suggestion. The definition of ECE was provided in Appendix B.2. Based on the suggestion, we added a reference to the main text in [L222].
>
> > 4. “How is the value $M$ determined in Eq. (1) and subsequent experiments?”
>
> The number of inducing points is set to 200 in our experiments as stated in [Supplementary, L45]. We provided additional experiments on performance sensitivity with respect to the number of inducing points in Appendix C.4-C.8. In general, classification performance can be improved by increasing the number of inducing points because of the increasing representation power. However, it requires a higher computational cost. Thus, we set it to 200 which works well in our experiments. .
>
> > 5. “In algorithm 1 step 3, how is $q(f_v)$ computed by Eq. (7)? Is there an analytical solution to the integration?”
>
> There is an analytical solution for Equation (7) because both the conditional prior and the marginal variational distribution are Gaussian distributions. From Equation (1), we can write the conditional prior as $p (\mathbf f_v \vert \mathbf u_v ) = \mathcal N (\mathbf K_{NM} \mathbf K_{MM}^{-1} \mathbf u_v, \mathbf K_{NN}-\mathbf K_{NM} \mathbf K_{MM}^{-1} \mathbf K_{NM}^{T})$, and we defined the marginal variational distribution in [L112] as $q(\mathbf u_v)=\mathcal N (\mathbf m_v, \mathbf S_v)$ . By using Gaussian linear transformation and integrating $\mathbf u_v$ out, we can derive the analytical solution as follows:
> \begin{align}
> q(\mathbf f_v) & \coloneqq \int p(\mathbf f_v \vert \mathbf u_v)q(\mathbf u_v) \\,d \mathbf u_v \\\\
> &=\int \mathcal N (\mathbf f_v;\mathbf K_{NM}\mathbf K_{MM}^{-1} \mathbf u_v, \mathbf K_{NN}-\mathbf K_{NM}\mathbf K_{MM}^{-1}\mathbf K_{NM}^{T}) \mathcal N (\mathbf u_v;\mathbf m_v,\mathbf S_v) \\,d\mathbf u_v \\\\
> &=\int \mathcal N (\mathbf f_v;\mathbf K_{NM}\mathbf K_{MM}^{-1}\mathbf m_v,\mathbf K_{NM}\mathbf K_{MM}^{-1}\mathbf S_v(\mathbf K_{NM}\mathbf K_{MM}^{-1})^{T}+ \mathbf K_{NN}-\mathbf K_{NM}\mathbf K_{MM}^{-1}\mathbf K_{NM}^{T})\mathcal N (\mathbf u_v;\mathbf m_v,\mathbf S_v) \\,d\mathbf u_v \\\\
> &=\mathcal N (\mathbf f_v;\mathbf K_{NM}\mathbf K_{MM}^{-1}\mathbf m_v,\mathbf K_{NM}\mathbf K_{MM}^{-1}\mathbf S_v(\mathbf K_{NM}\mathbf K_{MM}^{-1})^{T}+ \mathbf K_{NN}-\mathbf K_{NM}\mathbf K_{MM}^{-1}\mathbf K_{NM}^{T}) \int \mathcal N (\mathbf u_v;\mathbf m_v,\mathbf S_v) \\,d\mathbf u_v \\\\
> &=\mathcal N (\mathbf f_v;\mathbf K_{NM}\mathbf K_{MM}^{-1}\mathbf m_v,\mathbf K_{NM}\mathbf K_{MM}^{-1}\mathbf S_v(\mathbf K_{NM}\mathbf K_{MM}^{-1})^{T}+ \mathbf K_{NN}-\mathbf K_{NM}\mathbf K_{MM}^{-1}\mathbf K_{NM}^{T})
> \end{align}

---

> > ### Author Response · Authors · 2022-08-02
> > **Response (2/3)**
> >
> > > 6. “In OOD experiment, how is the three views used for training selected? Can we expect similar results if we attempt training and evaluation with a different choice of views such as in a manner of k-fold cross validation?”
> >
> > Ideally, one may conduct a cross-validation on every combination of choosing three views out of 15 types of corrupted views. However, this requires $\binom{15}{3}=455$ combinations where each combination has 50,000 training samples with 3 views. In the OOD experiment, we simply selected the first three corruptions, which are the same for all the compared methods.
> >
> > > 7. “Can the authors provide some discussion on the computation cost vs. accuracy given Monte Carlo estimation is used for Eq. (14) during inference?”
> >
> > Thank you for this suggestion. We conducted additional experiments to see how the number of Monte Carlo samples for Equation (14) affects the test accuracy.
> >
> > $\begin{array}{cccccccc}  \\# \text{Monte Carlo Samples} & \text{Handwritten} & \text{CUB} & \text{PIE} & \text{Caltech101} & \text{Scene15} & \text{HMDB} & \text{CIFAR10-C} \\\ \hline 1 & 98.25\pm0.40 & 85.50\pm2.09 & 83.38\pm2.47 & 91.16\pm0.40 & 65.55\pm1.10 & 66.38\pm0.86 & 70.86\pm0.36 \\\ 10 & 98.50\pm0.25 & 91.33\pm2.09 & 90.29\pm1.90 & 92.86\pm0.47 & 69.95\pm1.50 & 71.88\pm0.72 & 73.01\pm0.14 \\\  25 & 98.45\pm0.11 & 92.50\pm1.02 &   91.62\pm1.12 &   93.00\pm0.25&   69.75\pm1.16 &   71.95\pm0.43 & 73.15\pm0.12 \\\ 50 & 98.60\pm0.14 & 92.33\pm0.37 &   91.91\pm0.52 &   93.06\pm0.33 &   69.80\pm0.73 &   72.37\pm0.30 & 73.27\pm0.12 \\\  75 & 98.50\pm0.18 & 92.33\pm1.37 &   91.91\pm0.52 &   93.08\pm0.20 &   70.42\pm0.37 &   72.48\pm0.07 & 73.20\pm0.19 \\\ 100 & 98.60\pm0.14 & 92.33\pm0.70&   92.06\pm0.96&   93.00\pm0.33&   70.00\pm0.53&   72.30\pm0.19 & 73.30\pm0.05 \\\ 125 & 98.60\pm0.14 & 92.50\pm1.18 & 91.62\pm0.40 & 93.06\pm0.31 & 70.18\pm0.53 & 72.50\pm0.33 & 73.28\pm0.04 \end{array}$
> >
> > Although we can see improvements in accuracy as the number of Monte Carlo samples increases from 1 to 10 across all the datasets, there is no significant difference when the number of samples becomes large, like 75 vs 100 vs 125. In our experiments, we fixed it to 100 samples.
> >
> > > 8. “In Figure 3, the performance of TMC appears to be comparable to DE (EF) but worse than DE (LF), which is different from what reported in [10], which also had a much wider range of noise magnitude. Can the authors comment on what may contribute to this difference?”
> >
> > The performance difference is due to three differences between the setting in [11] and our experimental setting.
> > First, [11] has not implemented DE(LF) in their baseline. The reported deep ensemble in [11] is equivalent to DE(EF) in our setting where the multi-view input is concatenated as single-view. For DE(LF), on the other hand, we trained individual classifiers for each view separately and averaged the predictions.
> >
> > Second, [11] introduced the Gaussian noise and then normalized the noisy input by taking the maximum of the noisy input, whereas we normalized the input first and then introduced the Gaussian noise where the result is shown in Figure 3 (main text). Because the normalized input ranges from 0 to 1, we found noise std of 0.01 to 10 was sufficient. Given that the range of raw data of each view is significantly different (e.g., view 1’s input=[0,1] and view 2’s input=[0,10000]), we preferred to normalize the input first (i.e., our current setting) to ensure the equal impact of noise to every view. For the completion of our experiments of the comparison with TMC [11], in addition to Figure 3 (main text), we also reported the experimental results implemented by the setting of [11] in our Appendix (Figure 4) with a wider range of noise from 0.01 to 10,000. In this setting, ours still outperforms TMC.
> >
> > Third, both our work and [11] selected half of the views to introduce the Gaussian noise, but [11] has not explicitly described nor shown in their released code how the noisy views were selected. In our setting we computed every combination of choosing half from all of the views and averaged the results over all the combinations, which we believe is more comprehensive than [11].
> >
> > > 9. “Typo: Line 69 'applyinh' should be 'applying'”
> >
> > Thank you for pointing out the typo. We have revised it.

---

> > > ### Author Response · Authors · 2022-08-02
> > > **Response (3/3)**
> > >
> > >
> > > ***Additional references***
> > >
> > > [R1] M. Abdar, F. Pourpanah, S. Hussain, D. Rezazadegan, L. Liu, M. Ghavamzadeh, P. Fieguth, X. Cao, A. Khosravi, U. R. Acharya, V. Makarenkov, and S. Nahavandi. A review of uncertainty quantification in deep learning: Techniques, applications and challenges. *Information Fusion*, 76:243–297, 2021. ISSN 1566-2535. doi: https://doi.org/10.1016/j.inffus.2021.05.008.
> > >
> > > [R2] Y. Ovadia, E. Fertig, J. Ren, Z. Nado, D. Sculley, S. Nowozin, J. Dillon, B. Lakshminarayanan, and J. Snoek. Can you trust your model's uncertainty? evaluating predictive uncertainty under dataset shift. In H. Wallach, H. Larochelle, A. Beygelzimer, F. d'Alché-Buc, E. Fox, and R. Garnett, editors, *Advances in Neural Information Processing Systems,* volume 32. Curran Associates, Inc., 2019.
> > >
> > > [R3] J. Mukhoti, A. Kirsch, J. van Amersfoort, P. H. Torr, and Y. Gal. Deterministic neural networks with appropriate inductive biases capture epistemic and aleatoric uncertainty. *arXiv preprint arXiv:2102.11582*, 2021.

---

> > > > ### Comment · Reviewer_2rPi · 2022-08-10
> > > > **Final Evaluation**
> > > >
> > > > Thanks the authors for a comprehensive rebuttal. Most of my concerns are addressed. I think some part of the rebuttal should be incorporated into the final revision such as answers to point 1, 2, 5, and 8 to better reveal some important technical details and justify the significance of the results. Regarding my point 6, I don't expect exhaust all combinations. But it will strengthen the results even if the authors just do a 5-fold evaluation for example. For my point 7, the authors did not respond to the total inference time vs. number of Monte Carlo samples. Overall, the rebuttal is still quite thorough and many additional experiments are already included in supplementary materials. I'm willing to upgrade my rating by one level.

---

### Official Review · Reviewer_7qYz · 2022-07-12

**Rating:** 8
**Confidence:** 5
**Soundness:** 4 excellent
**Presentation:** 4 excellent
**Contribution:** 4 excellent

**Summary:**

This paper is about uncertainty quantification for multi-view inputs, where multiple views are input to a model and a decision (classification or regression) must be made by combining decisions made with each individual input/view.

The authors propose a new kind of Gaussian Process, the Multi-View GP, that works by combining predictions from individual views with a Product of Experts (PoE) and training/estimating the posterior predictive distribution using variationa inference. The main purpose seems to be robustness to noise in some of the views, and out of distribution detection.

The contributions of this work are:
- A concept for uncertainty quantification on multi-view data using Gaussian Processes and Variational Inference.
- The proposed framework improves on the state of the art for multi-view/multi-modality classification under noise corrupted inputs, calibration, and out of distribution detection, on several datasets.
- Multi-view posterior distributions are approximated using variational inference in a theoretically grounded way, and predictions for each view are inversely weighted to their uncertainty, meaning that only the most certain views contribute the most to the final output of the model.

**Questions:**

- How was Product of Experts chosen? [27] shows that there are other options, like Mixture of Experts, and Naive Local Experts, so was there an ablation using these other choices to combine individual views?
- Would it be possible to also use Deep Ensembles (+ some approximate Bayesian NN) instead of Gaussian Processes for each of the views, while keeping the variational posterior distribution?

**Limitations:**

The supplementary material contains a nice section on societal impacts, mostly about the approximation to uncertainty which could be misleading, and  to raise awareness of the importance of uncertainty quantification for safety and trustability of AI systems.

There are some obvious limitations that could be mentioned or discussed, such as that it provides an approximation (due to approximate GP, variational inference, etc), and that the formulation is made for classification. The authors mention that the weight computation can be suboptimal, which I believe is true.


**Strengths And Weaknesses:**

Strengths
- The paper deals with the problem of multi-view/multi-modal inputs to machine learning models, in particular Gaussian Processes, combined with uncertainty quantification. As the authors say, this specific combination (multi-view + uncertainty) is often not well explored. I believe the ideas presented in this paper are novel, in particular the use of Gaussian Processes for multi-view/multi-modal data, and the results show that this combination is robust to noise.
- This work is very well written, the ideas flow very clearly and it was a pleasure to read. I only have some minor comments below.
- The formulation used in the paper makes sense. First there is one Gaussian Process for each input view/modality, and the predictions are combined using Product of Experts. The joint predictive distribution is approximated using variational inference, in a very standard framework by maximizing the ELBO. At prediction time, the different views/modalities are weighted by the negative entropy of the predictive distribution, which gives more weight to more certain views/modalities. The whole formulation is very well theoretically grounded and connected to the literature.
- Overall the baselines selected are appropriate for comparison. I have no doubts about the evaluation, but in Weaknesses I mention some additional baselines that could be a good comparison.
- One experiment uses 6 common multi-view datasets, and adds Gaussian noise to half of the views, simulating a case of corrupt data or sensor failure. In this experiment, the proposed method MGP clearly outperforms all baselines in terms of accuracy as noise is increased, but more notably MGP seems to be very robust to noise in some of the views, with minor drops in accuracy as the noise standard deviation is increased.
- Another experiment tests out of distribution detection in a synthetic multi-view dataset based on CIFAR10-C, where views are different corruptions of the dataset, and SVHN/CIFAR100 are used as out of distribution datasets. In this setup, for the in-distribution data, MGP has the best calibration error, no significant degradation in accuracy, and best accuracy when corrupting half views with Gaussian Noise. For the OOD detection setting, MGP has the best AUC.
- The paper has a very nice and detailed supplementary material with proofs and additional results and details.
- Overall I believe that this paper is well executed, the ideas are novel, and the results show that it is a very good improvement over the state of the art. I believe the paper should be accepted.

Weaknesses
- One major weakness is that the proposed method seems only to be defined for classification, but I believe it is possible to extend it to regression, but the paper does not evaluate or describe the formulation for a regression setting.

~~- I think in terms of baselines, some methods like deep ensembles could also  be trained in a similar way to weight each view prediction with the negative entropy, in order to show if this mechanism is important to improve performance, particularly in noisy views. A deep ensemble could also in theory be combined with Bayes by Backprop and be trained using the Variational approximation, not sure if this would work but the authors could mention if this is viable and why it was not tried.~~

Minor issues
- In Fig 1 and 2, it would be nice to have a proper legend for the colorbar on the right, indicating what exactly each heatmap is representing.
- I think the paper is using the ECE metric without defining it, I am aware it is the Expected Calibration Error, and is clearly defined in the supplementary, but some indication of what it is, like a reference, should be in the main paper.

---

> ### Author Response · Authors · 2022-08-01
> **Response (1/2)**
>
> We would like to thank the reviewer for such comprehensive and constructive review. We especially appreciate the reviewer’s detailed and thorough comments on the strengths and insightful feedback on the weaknesses. We would like to address the reviewer’s comments and questions as follows:
>
> > 1. “...the paper does not evaluate or describe the formulation for a regression setting.”
>
> Thanks for an insightful comment. We totally agree with the reviewer. The proposed framework can be easily extended to a regression setting. We can skip the reparameterization of class labels to regression labels which is Equation (5) and model a Gaussian likelihood with homoscedastic noise variance as $p(y|\mathbf{f}_v)=\mathcal{N}(\mathbf{f}_v, \sigma_n I)$ instead of the heteroscedastic noise variance used in the paper. Then, the noise variance $\sigma_n$ can be optimized as a learnable parameter. In this work, however, we focus on classification tasks as we aim to apply the proposed framework in a real-world medical setting with classification labels.
>
> > 2. “...deep ensembles could also be trained in a similar way to weight each view prediction with the negative entropy...”
>
> We appreciate the reviewer's suggestion and have added another baseline. To be precise, we estimated the entropy of prediction $H(p_v)=-\sum_{c=1}^C p_{v,c}\log(p_{v,c})$ for each view’s prediction $p_v$ of DE(LF) where $v=1, ..., V$ and $c=1, ..., C$. Then, the prediction is weighted by negative entropy that is normalized by $\alpha_v=\frac{exp(-H(p_v))}{\sum_{k=1}^V exp(-H(p_k))}$. The combined DE (LF)’s prediction is thus $p_{DE(LF)+\alpha_v}=\sum_{v=1}^V \alpha_v p_v$. The updated results are as follows:
>
> In-domain test accuracy $\uparrow$
>
> $\begin{array}{ccccccc} \text{Method} & \text{Handwritten} & \text{CUB} & \text{PIE} & \text{Caltech101} & \text{Scene15} & \text{HMDB} \\\ \hline DE (LF)    &   99.25\pm0.00 &   92.33\pm0.70 &   87.21\pm0.66 &   92.97\pm0.13 &   67.05\pm0.38 &   69.98\pm0.36 \\\ DE (LF)+\alpha_v & 99.25\pm0.00 & 92.83\pm0.95 & 87.79\pm0.40 & 92.35\pm0.12 & 67.30\pm0.39 & 69.70\pm0.30 \end{array}$
>
> In-domain test ECE $\downarrow$
>
> $\begin{array}{ccccccc} \text{Method} & \text{Handwritten} & \text{CUB} & \text{PIE} & \text{Caltech101} & \text{Scene15} & \text{HMDB} \\\ \hline DE (LF)   &   0.292\pm0.001&   0.270\pm0.009&   0.567\pm0.006&   0.023\pm0.002&   0.319\pm0.005&   0.270\pm0.003 \\\ DE (LF)+\alpha_v & 0.157\pm0.001 & 0.193\pm0.014 & 0.367\pm0.004 & 0.025\pm0.001 & 0.268\pm0.005 & 0.207\pm0.003  \end{array}$
>
> Average test accuracy with Gaussian noise (std from 0.01 to 10) added to half of the views
>
> $\begin{array}{ccccccc}  \text{Method} & \text{Handwritten} & \text{CUB} & \text{PIE} & \text{Caltech101} & \text{Scene15} & \text{HMDB} \\\ \hline DE (LF)  & 95.63\pm0.08&   76.16\pm0.28&   67.69\pm0.35&   81.85\pm0.14&  50.13\pm0.27&   43.01\pm0.19 \\\  DE (LF)+\alpha_v & 82.64\pm0.10 & 75.49\pm0.80 & 62.05\pm0.51 & 62.05\pm0.51 & 44.74\pm0.24 & 40.65\pm0.17   \end{array}$
>
> Out-of-domain detection results with CIFAR10-C
>
> $\begin{array}{ccccc}  \text{Method} & \text{Test accuracy} & \text{Test ECE} & \text{OOD AUROC (SVHN)} & \text{OOD AUROC (CIFAR100)} \\\ \hline DE (LF) & 75.40\pm0.06 & 0.095\pm0.001 & 0.722\pm0.016 & 0.693\pm0.006 \\\  DE (LF)+\alpha_v & 75.32\pm0.09 & 0.048\pm0.001 & 0.722\pm0.016 & 0.693\pm0.006   \end{array}$
>
> When “$DE(LF)$” and the newly added “$DE(LF)+\alpha_v$” are compared, in-domain test accuracy and in-domain test ECE are improved by adding the weight term in most cases. However, the robustness to noise is heavily compromised. This result is not surprising because when the testing domain is shifted from the training domain, we showed that $DE(LF)$’s predictions become overconfident as shown in Figure 2 (a)-(e). Thus, the predictive entropy cannot correctly capture the domain shift and becomes less meaningful. Similarly, no improvement of OOD detection was observed. Nevertheless, we thank the reviewer for the suggestion.
>
>
> > 3. “A deep ensemble could also in theory be combined with Bayes by Backprop and be trained using the Variational approximation...”
>
> We agree with the reviewer that the Deep Ensemble [23] (DE) can be theoretically combined with the Bayes by Backprop as a DE approximates the distributions of a Bayesian neural network’s weights and predictions by deterministic ensemble members. However, as a DE requires training independent ensemble members in parallel [23], we believe that it is not straightforward to directly apply the Bayes by Backprop to DE. It is also possible to build an ensemble of Bayesian neural networks with the Bayes by Backprop. However, it has not been considered in our study as it requires even higher computational and memory cost than a DE or a Bayesian neural network, which may not be suitable for the practical setting in our medical project.

---

> > ### Author Response · Authors · 2022-08-01
> > **Response (2/2)**
> >
> > >4. “How was Product of Experts chosen? [27] shows that there are other options, like Mixture of Experts, and Naive Local Experts, so was there an ablation using these other choices to combine individual views?”
> >
> > Thank you for raising this important question. The naive local experts partition a training dataset into multiple subsets and train individual local experts at each subset [R1]. Given a testing sample close to a certain training subset, the allocated single expert makes the prediction (i.e. a single representative expert makes predictions). In a multi-view setting, however, this would disregard meaningful predictions from different views if only a single view/expert is selected to make predictions. The mixture of experts [R2] can be extended to multi-view data by introducing a gating function that weights each view’s posterior. However, the mixture of experts cannot be sharper than the sharpest expert because it is the weighted sum of experts [17,28]. Even if we have a very accurate expert, the overall quality of predictions degrades. Product of experts [6], on the other hand, can be sharper than the sharpest expert because of the multiplication to combine the distributions. This is a desirable property especially when some views are noisy or corrupted in order to maintain high accuracy from clean or non-corrupted views. Nevertheless, we would like to investigate empirical differences between the mixture of experts and the product of experts in our future study.
> >
> > > 5. “Would it be possible to also use Deep Ensembles (+ some approximate Bayesian NN) instead of Gaussian Processes for each of the views, while keeping the variational posterior distribution?”
> >
> > We believe it would be possible to use Deep Ensembles (+ some approximate Bayesian NN) instead of Gaussian Processes for each of the views, while keeping the variational posterior distribution as our framework. Our framework is not limited to GPs if the variational posterior is obtainable. However, we believe that GP is a more suitable method for robustness to noise and OOD detection experiments because a GP is a non-parametric method which inherently measures the correlation between training samples and testing samples via a kernel function in the covariance matrix [49]. If the kernel function is chosen properly, it can properly measure the dot product within a high dimensional space (the reproducing kernel Hilbert space). It has been also shown in [29] that a GP outperforms other baselines such as a deep ensemble with a single-view setting. Nonetheless, it will be very interesting to see how other methods such as Bayesian neural networks would perform with our proposed method. We will leave this in our future work.
> >
> > > 6. ECE metric and labels of colorbar in Figure 1 and 2.
> >
> > Based on the reviewer’s suggestions, we added a reference of ECE to the main text in [L222] and added labels of colorbars in Figure 1 and 2.
> >
> > **Additional references**
> >
> > [R1] H.-M. Kim, B. K. Mallick, and C. C. Holmes. Analyzing nonstationary spatial data using piecewise gaussian processes. *Journal of the American Statistical Association*, 100(470): 653–668, 2005. doi: 10.1198/016214504000002014.
> >
> > [R2] C. Rasmussen and Z. Ghahramani. Infinite mixtures of gaussian process experts. In T. Dietterich, S. Becker, and Z. Ghahramani, editors, *Advances in Neural Information Processing Systems*, volume 14. MIT Press, 2001.

---

> > > ### Comment · Reviewer_7qYz · 2022-08-10
> > > **Great response**
> > >
> > > Thank you for the detailed response, I think my questions and concerns have been addressed.

---

### Official Review · Reviewer_LsN2 · 2022-07-12

**Rating:** 3
**Confidence:** 4
**Soundness:** 1 poor
**Presentation:** 2 fair
**Contribution:** 2 fair

**Summary:**

This paper proposes a new uncertainty estimation framework with Gaussian Process for multi-view/modal data. The framework is basically a product-of-expert model. The model is trained via a variational inference algorithm to approximate the multi-view posterior distributions. Experiments on both synthetic and real-worl data are conducted.

**Questions:**

The whole part of methodolgy needs to be clarified and improved. Please see the problems in Weaknesses.

**Limitations:**

The limitations and potential negative societal impact are not explained in the paper.

**Strengths And Weaknesses:**

# Strengths


# Weaknesses
1. The paper is hard-to-follow, especially in Sec. 2. [L63] The latent function $f_v$ is not defined. Inferring from the context, the dimension of $f_v$ is $N$, i.e., the number of training samples. However, in many real-world dataset, $N$ can be ten thousand to one million, in which case the covariance is intractable. [L66] The kernel function indicates that the inputs $x_{v,i}$ are real numbers, while in practice the inputs are often high dimensional images or discrete tokens (like captions). [L114] What is the $c$ in $\tilde{y}_{i, c}$? After reading the whole section, the reviewer cannot find the origin of **Gaussian Process** and thinks the term is inappropriate.
2. The experiments are not convictive. There is no metrics for synthetic dataset. The rationality of synthetic dataset needs to be justified. Why can the two moon curves be considered as two views of the samples from the same class? Why are the red and blue classes not symmetric?

Typos:
[title] Uncertainty Estimation for Multi-View Data
[L69] applyinh -> applying

---

> ### Author Response · Authors · 2022-08-01
> **Response**
>
> Thank you for your comments. We would like to address the following questions.
>
> > 1. “The latent function $\mathbf{f}_v$ is not defined.”
>
> By definition, a Gaussian process (GP) is a stochastic process with Gaussian distributions over functions. The latent function is not defined with parametric forms. The GP is defined using a Gaussian distribution with mean and the covariance matrix where its random variable is our latent function. We applied the standard procedure of building and learning GPs [49] where the prior of the GP has a zero mean matrix and $\mathbf{K}_{NN}$ as the covariance matrix [L63], and the likelihood and posterior are shown in [L92] and [L110] respectively.
>
> > 2. “In many real-world dataset, $N$ can be ten thousand to one million, in which case the covariance is intractable.”
>
> Originally, GP requires $\mathcal{O}(N^3)$ to compute the inversion of the covariance matrix. Extensive works have been conducted to overcome this limitation, and in our paper, we use sparse GPs [28], where $M$ inducing points ($M<<N$) are introduced. This significantly reduces the computational cost to $\mathcal{O}(M^3)$ in [L76]. In our experiments, we set $M$ to $200$, which works well in practice, and we provide the performance sensitivity with respect to $M$ in Appendix C.4-8.
>
> > 3. “The kernel function indicates that the inputs $x_{v,i}$ are real numbers, while in practice the inputs are often high dimensional images or discrete tokens (like captions).”
>
> Our multi-view GP works in the low-dimensional feature space extracted by feature extractors, instead of the high-dimensional data space. That is how GPs can handle high dimensional data [29]. We used a deep residual network and the Inception v3 [39] for feature extractors for the synthetic dataset experiment and the OOD samples detection respectively, as stated in [Supplementary, L30]  and [L237]. Similarly, the dataset used for the robustness to noise experiment is a feature set [11].
>
> > 4. “What is the $c$ in $\tilde{y}_{i,c}$?”
>
> We acknowledge that $c$ is implicitly defined in [L94]. We would like to clarify that $c$ is the index of class ranging from $1$ to $C$, where the total number of classes is $C$ [L62].
>
> > 5. “After reading the whole section, the reviewer cannot find the origin of **Gaussian Process** and thinks the term is inappropriate.”
>
> Our work introduces the sparse variational GP [14,28] for each view, which is described in Section 2.1. Then, we proposed a principled way to combine the predictive posterior distributions by product of experts in Section 2.2. We follow the standard way of formulating GPs as found in [14,28,49] and variational inference to approximate the posterior with formulations aligned with the literature [4,14]. In the camera-ready version, we will provide more background of Gaussian processes to improve the clarity for readers who may not be familiar with GPs.
>
> > 6. “There is no metrics for synthetic dataset. The rationality of synthetic dataset needs to be justified.”
>
> The purpose of the synthetic dataset experiment is to demonstrate how the predictive behavior of our framework is distinct from those of baselines when test samples are far from training samples. Our synthetic dataset was extended from the widely used Moon dataset, which we believe is reasonable to demonstrate the differences of the behaviors of the compared methods [29,43,44].
>
> We believe that the visualizations on the synthetic dataset in Figure 1 and 2 are more convincing than numbers because inputs and outputs of models can be easily plotted in a 2D space. In terms of metrics, extensive experiments were done on a large number of real-world datasets with several key metrics of uncertainty estimation such as expected calibration error (ECE) and OOD AUROC [11,29,43,44].
>
> > 7. “Why can the two moon curves be considered as two views of the samples from the same class? Why are the red and blue classes not symmetric?”
>
> We generated different views of the moon dataset with the exactly same underlying function and the same number of points for each class except the different radius of circles. For instance, given the same $\theta=[0,\pi]$, we generate $(x,y)$ with different radius by $x=radius \times cos\theta$ and $y=radius \times sin\theta$. The moon dataset is a publicly available dataset from Scikit-learn ([Link](https://scikit-learn.org/stable/modules/generated/sklearn.datasets.make_moons.html)) which is often used to illustrate the predictive behavior as in [29,43,44].
>
> > 8. “The limitations and potential negative societal impact are not explained in the paper.”
>
> Our limitation regarding the sub-optimal choice of introducing the weight to balance the predictive posterior distribution during inference was outlined in [L256]. Also, we included potential negative societal impact in Appendix D that is the undue trust of estimated uncertainty could be possibly misleading.

---

> > ### Author Response · Authors · 2022-08-09
> > **Dear Reviewer**
> >
> > Dear Reviewer,
> >
> > Thanks again for the review. We hope our response has addressed your concerns.
> > If you have more questions, please let us know. We are more than happy to discuss it.
> > We wish you could kindly take our response and other reviewers' opinions into consideration.
> >
> > Best regards,
> >
> > The Authors

---

### Meta-Review · Area_Chair_Moa1 · 2022-08-25

**Recommendation:** Accept
**Confidence:** Certain

**Metareview:**

Two out of three reviewers learn to accept, with one of them championing the paper. The reviewer that’s most critical did not engage with either the authors or with the reviewers / AC during the discussion period. Furthermore, the review itself is pretty sparse in details and justifications on the criticism. Upon looking at the manuscript I agree with the other two positive reviewers that the work is innovative and the idea of combining multiple views under Product of Experts framework is interesting. Using Sparse GPs as an underlying mechanism does address some of the concerns about computational complexity of GPs. Additionally, the experimental evaluation is detailed enough and highlights the potential of the proposed model. Given all the above and including the observation that the manuscript is well-written with a principled inference mechanism, I am recommending the manuscript of acceptance at NeurIPS.

**Award:**

No

---

### Decision · Program_Chairs · 2022-09-14

Accept